# Balancing Performance and Costs in Best Arm Identification

**Michael O. Harding**
Department of Statistics
University of Wisconsin-Madison
moharding@wisc.edu

**Kirthevasan Kandasamy**
Department of Computer Science
University of Wisconsin-Madison
kandasamy@cs.wisc.edu

## Abstract

We consider the problem of identifying the best arm in a multi-armed bandit model. Despite a wealth of literature in the traditional fixed budget and fixed confidence regimes of the best arm identification problem, it still remains a mystery to most practitioners as to how to choose an approach and corresponding budget or confidence parameter. We propose a new formalism to avoid this dilemma altogether by minimizing a risk functional which explicitly balances the performance of the recommended arm and the cost incurred by learning this arm. In this framework, a cost is incurred for each observation during the sampling phase, and upon recommending an arm, a performance penalty is incurred for identifying a suboptimal arm. The learner's goal is to minimize the sum of the penalty and cost. This new regime mirrors the priorities of many practitioners, e.g. maximizing profit in an A/B testing framework, better than classical fixed budget or confidence settings. We derive theoretical lower bounds for the risk of each of two choices for the performance penalty, the probability of misidentification and the simple regret, and propose an algorithm called DBCARE to match these lower bounds up to polylog factors on nearly all problem instances. We then demonstrate the performance of DBCARE on a number of simulated models, comparing to fixed budget and confidence algorithms to show the shortfalls of existing BAI paradigms on this problem.

## 1 Introduction

Best Arm Identification (BAI) in multi-armed bandits is a fundamental problem in decision-making under uncertainty. The objective is to identify the arm with the highest expected reward by adaptively drawing samples from distributions associated with each arm. BAI arises in many real-world applications. In advertising, arms represent different ads, and the aim is to find the ad which maximizes revenue generated [18]. In statistical model selection, arms represent different hyperparameter configurations, and the aim is to find the best-performing one with minimal computational resources [20].

Traditionally, BAI has been studied under two paradigms: the *fixed budget* setting [2, 8], which seeks to maximize performance—i.e. the ability of a policy to recover the optimal arm—within a given sampling budget, and the *fixed performance* (e.g., fixed confidence [33, 15]) setting, which aims to minimize the number of samples needed to meet a target performance level. While algorithms for these settings have been successfully deployed in many real-world settings [32, 35, 51, 18], these settings are not a natural fit for all use cases. For instance, while determining the best arm is desirable, a slightly suboptimal choice may be acceptable if the cost of distinguishing between top candidates is prohibitively high. On the other hand, it is often unnecessary to continue sampling until reaching some pre-specified horizon when there is already enough evidence to determine the optimal arm.

To this end, we propose a novel paradigm for BAI, in which a policy should explicitly balance performance and sampling cost on the fly, without being constrained by a fixed performance level or a pre-specified sampling budget. This framework allows policies to *adaptively* terminate according to the difficulty of the problem. The following is an example where such a framework would be natural.

39th Conference on Neural Information Processing Systems (NeurIPS 2025).

**Example** (Advertising). *Consider a firm choosing among $K$ versions of an ad. To inform its choice, the firm may show versions to participants in a focus group (arm pull), incurring a cost $c$ per showing. The firm wishes to choose an algorithm to maximize the expected profit, i.e. the expected revenue of the selected ad $(\widehat{I})$ minus the expected cost of the sampling procedure: $\mathbb{E}[revenue_{\widehat{I}}] - c\,\mathbb{E}[\# arm\ pulls]$. Letting $I^\star$ be the ad with the highest expected revenue, then maximizing expected profit can be equivalently stated as minimizing $\mathbb{E}[revenue_{I^\star} - revenue_{\widehat{I}}] + c\,\mathbb{E}[\# arm\ pulls]$. Traditional fixed budget or confidence algorithms would be a poor fit for this problem, as it is unclear how one should choose the budget or confidence level to optimize the objective.*

## 1.1 Model

We will now formally introduce our setting. A learner has access to a MAB model $\nu = \{\nu_a\}_{a\in[K]}$, which consists of $K$ arms, each associated with a probability distribution $\nu_a$. Let $\mu_a = \mathbb{E}_{\nu_a}[X]$ denote the expected reward of arm $a$. Following common conventions in the BAI literature, we assume without loss of generality that the arms are ordered so that $\mu_1 \geq \mu_2 \geq \cdots \geq \mu_K$ (the learner is unaware of this ordering). We will assume that for each arm $a \in [K]$, the distribution $\nu_a$ is $\sigma$-sub-Gaussian and that $\mu_a \in [0, B]$. The learner is aware of $\sigma$ and $B$.

A learner interacts with the bandit model over a sequence of rounds $t = 1, 2, \ldots$. On round $t$, the learner selects an arm $A_t \in [K]$ according to a policy $\pi$ and observes an independent sample $X_t$ drawn from $\nu_{A_t}$. The choice of $A_t$ may depend on the history $\{(A_s, X_s)\}_{s=1}^{t-1}$ of previous actions and observations. Upon termination, the policy recommends an arm $\widehat{I} \in [K]$ as the estimated best arm.

**Prior work.** Traditionally, BAI has been studied under two main regimes: *(1) Fixed budget:* The learner is allowed at most $T \in \mathbb{N}$ samples and aims to minimize either the *probability of misidentification* [2] $\mathbb{P}(\mu_1 \neq \mu_{\widehat{I}})$ or the *simple regret* [8] $\mathbb{E}[\mu_1 - \mu_{\widehat{I}}]$, i.e. the expected gap between the optimal and selected arms. *(2) Fixed performance:* The learner must satisfy a specified performance goal while minimizing the number of samples. The most common instantiation is *fixed-confidence* BAI [6, 16], where the probability of misidentification $\mathbb{P}(\mu_1 \neq \mu_{\widehat{I}})$ is at most a given goal $\delta$.

**This work.** Both the fixed-budget and fixed-performance formulations neglect practical situations where one may not have a pre-specified budget or performance goal, but would like to trade-off between performance and sampling cost based on problem difficulty. Motivated by such considerations, we propose a new setting, where the goal is to minimize a risk functional that captures both a performance penalty and the cumulative sampling cost. Choosing either the probability of misidentification or the simple regret as the penalty, we study the following two risk measures:

$$
\begin{aligned}
\mathcal{R}_{\mathrm{MI}}(\pi, \nu) &:= \mathbb{E}_{\nu,\pi}\left[\mathbb{1}\left(\mu_1 \neq \mu_{\widehat{I}}\right) + c\tau\right] = \mathbb{P}_{\nu,\pi}\left(\mu_1 \neq \mu_{\widehat{I}}\right) + c\,\mathbb{E}_{\nu,\pi}[\tau], \\
\mathcal{R}_{\mathrm{SR}}(\pi, \nu) &:= \mathbb{E}_{\nu,\pi}\left[\left(\mu_1 - \mu_{\widehat{I}}\right) + c\tau\right] = \mathbb{E}_{\nu,\pi}\left[\mu_1 - \mu_{\widehat{I}}\right] + c\,\mathbb{E}_{\nu,\pi}[\tau].
\end{aligned}
\tag{1}
$$

Here, $c > 0$ is the (known) cost required to collect a sample, relative to the performance penalty, and $\tau$ is the stopping time (total number of samples) of the policy $\pi$. Moreover, $\mathbb{P}_{\nu,\pi}$ and $\mathbb{E}_{\nu,\pi}$ denote the probability and expectation with respect to all randomness arising from the interaction between policy $\pi$ and the bandit model $\nu$.

## 1.2 Summary of our contributions and results

**Novel problem formalism.** To the best of our knowledge, we are the first to study this risk-based formalism for BAI which trades off between performance and sampling costs. We design policies for both risk measures in (1), upper bound the risk, and provide nearly matching lower bounds.

**Lower bounds.** To summarize our lower bounds, let $\Delta_k = \mu_1 - \mu_k$ denote the sub-optimality gap of arm $k$, and let $H := \sum_{k=2}^K \Delta_k^{-2}$ be a problem complexity parameter [33, 15, 26, 21, 28, 30]. We show that the problem difficulty exhibits a phase transition depending on the magnitude of $H$ and the smallest gap $\Delta_2$. Specifically, in the case of $\mathcal{R}_{\mathrm{MI}}$, when $H \in \mathcal{O}((\sigma^2 c)^{-1})$, we show that $\mathcal{R}_{\mathrm{MI}} \in \Omega\left(c\sigma^2 H \log\left(((c\sigma^2 H)^{-1})\right)\right)$, and otherwise, $\mathcal{R}_{\mathrm{MI}} \in \Omega(1)$. In the case of $\mathcal{R}_{\mathrm{SR}}$, when $H\Delta_2^{-1} \in \mathcal{O}((\sigma^2 c)^{-1})$, we show that $\mathcal{R}_{\mathrm{SR}} \in \Omega\left(c\sigma^2 H \log\left(\Delta_2(c\sigma^2 H)^{-1}\right)\right)$, and otherwise, $\mathcal{R}_{\mathrm{SR}} \in \Omega(\Delta_2)$. This phase transition—absent in classical fixed-confidence or fixed-budget settings—underscores the trade-off between performance and costs inherent to our setting: probabilistically distinguishing sub-Gaussian arms scales inversely with the size of the gaps between them, so with small enough gaps it becomes optimal to simply guess the best arm without incurring the cost of sampling.

*Proof ideas.* Our proof employs change-of-measure arguments to lower bound the risk associated with any particular algorithm via an auxiliary function of problem parameters and the expected stopping time of the algorithm, $\mathbb{E}_{\nu,\pi}[\tau]$. Crucially, this function is convex in $\mathbb{E}_{\nu,\pi}[\tau]$, and minimizing it with respect to $\mathbb{E}_{\nu,\pi}[\tau]$ yields lower bounds on the performance of *any* algorithm while additionally revealing the phase transition behavior, via the regions where $\mathbb{E}_{\nu,\pi}[\tau] = 0$ is optimal.

**Algorithm.** We propose `DBCARE` (**D**ynamically **B**udgeted **C**ost-**A**dapted **R**isk-minimizing **E**limination) for this setting. `DBCARE` maintains a subset $S \subset [K]$ of surviving arms and confidence intervals for the mean values of these arms. It takes as input a function $N^\star : \mathbb{N} \to \mathbb{N}$ of the size of $S$, which determines the maximum number of times each arm in $S$ may be pulled. It proceeds in epochs, where in each epoch, every surviving arm is pulled once. At the end of each epoch, `DBCARE` eliminates arms that can be confidently identified as suboptimal based on the confidence intervals. If any arms are eliminated, the budget for each surviving arm is updated based on $N^\star$. If the budget of arm pulls is exhausted before there is a clear winner, i.e. only one surviving arm, it recommends the surviving arm with the highest empirical mean. However, if a clear winner emerges before the current budget, it terminates early and recommends this arm.

`DBCARE` combines ideas from both fixed-budget and fixed-confidence algorithms for BAI. However, unlike fixed budget algorithms, the budget is not given in advance; rather, the total number of times an arm can be pulled is determined by the function $N^\star$ which depends on the risk (1), the cost $c$, and the size of the current surviving set $S$. Similarly, unlike algorithms for fixed confidence BAI [24, 21], the confidence intervals are carefully chosen based on problem parameters, and not via a prespecified failure probability target $\delta$. This design allows `DBCARE` to adapt to the problem difficulty with respect to the gaps and cost, while simultaneously ensuring control over the worst-case risk.

**Upper bound.** We show that the above algorithm, with carefully chosen parameters, matches the lower bounds in almost all regimes. Specifically, for $\mathcal{R}_{\mathrm{MI}}$, our algorithm matches the lower bound up to polylog factors for all values of the complexity parameter $H$. For $\mathcal{R}_{\mathrm{SR}}$, we similarly match the lower bound up to polylog factors when $H$ is not too large. However, when $H \to \infty$, our upper bound scales as $\mathcal{O}(\log(K)(K\sigma^2 c)^{1/3})$, while the lower bound is $\Omega(\Delta_2)$, leaving an additive gap.

Despite this discrepancy in the $\mathcal{R}_{\mathrm{SR}}$ case, we make two important observations. First, we show that our algorithm is *minimax optimal*; that is, the worst-case risk over all problem instances matches the worst-case lower bound up to logarithmic factors. Second, the lower bound in the large $H$ regime is tight and cannot be improved: a naive guessing algorithm—one that selects an arm without pulling any—achieves the lower bound on certain problem instances in this region. However, such a policy performs poorly when $H$ is small, underscoring the value of our adaptive strategy.

*Proof ideas.* Our use of an elimination-style procedure allows us to guarantee that we never eliminate the optimal arm with high probability, and also identify precisely when highly suboptimal arms are guaranteed to be eliminated. Then, by choosing $N^\star(|S|) \asymp \mathcal{O}((|S|\,c)^{-1})$ for $\mathcal{R}_{\mathrm{MI}}$ and $N^\star(|S|) \asymp \mathcal{O}(\sigma^{2/3}(|S|\,c)^{-2/3})$ for $\mathcal{R}_{\mathrm{SR}}$, we ensure that `DBCARE` can both match the worst-case behavior of the lower bound and adapt to easier problem settings where there are relatively few good candidate arms.

**Empirical evaluation.** We corroborate these theoretical findings in simulations and in a real-world experiment on a drug discovery dataset. We compare to fixed budget and confidence algorithms to show the deficiencies of naive adaptations of existing BAI paradigms on this problem.

## 1.3 Related work

**BAI.** The multi-armed bandit (MAB) problem, first introduced by Thompson [45], has become a foundational framework for studying the exploration-exploitation trade-off in sequential decision-making under uncertainty. Within this framework, Best Arm Identification (BAI) focuses on identifying the arm with the highest expected reward [7, 25, 16, 10, 28, 21, 41].

BAI has primarily been studied under two paradigms: the fixed-budget and fixed-performance settings. In the fixed-budget setting, the objective is to minimize the probability of misidentification [2, 29, 30, 11, 4], or alternatively, to minimize the simple regret [7, 8, 52]. In the fixed performance setting, the majority of the literature has focused on achieving a target probability of misidentification (a.k.a fixed confidence BAI) [14, 33, 15, 21, 17, 24, 23]. To the best of our knowledge, there is no prior work on minimizing the number of pulls subject to a performance goal on the simple regret.

Our work builds on the extensive literature in this area. In particular, our algorithm draws inspiration from racing-style methods developed for fixed-confidence BAI [34, 21, 24], while our lower bounds rely on technical lemmas from Kaufmann et al. [30]. Nevertheless, the problem we study departs meaningfully from existing formulations, requiring new conceptual insights and analytical tools.

**Cost of arm pulls in MAB.** Several works have explored sampling costs in BAI. Xia et al. [49] and Qin et al. [39] study identifying the arm with highest reward-to-cost ratio, assuming both reward and cost are observed per sample, both in fixed-budget and fixed-confidence settings. In contrast, in our setting, once a final arm is selected, only its expected reward—not its sampling cost—remains relevant. Degenne et al. [13] and Yang et al. [50] consider minimizing the cumulative regret [40] while performing BAI, but this approach is not applicable when sampling costs are exogenous to rewards, as we consider in our setting. Kanarios et al. [27] study minimizing cumulative cost (instead of the number of pulls) in a fixed confidence setting, when the learner observes a stochastic cost on each arm pull in addition to the reward. Recent work in multi-Fidelity BAI [37, 48, 38] allows a learner to choose to incur different costs for varying magnitudes of accuracy. The last two problem settings are distinctly different from ours. Finally, some works [3, 43] address costs in the cumulative regret setting, which is also distinct from our focus on BAI.

**Bayesian sequential testing in classical statistics.** Arrow et al. [1] and Wald and Wolfowitz [47] study Bayesian formulations of sequential binary hypothesis testing problems (e.g., $H_1 : \mu_1 - \mu_2 = \Delta$ vs. $H_2 : \mu_1 - \mu_2 = -\Delta$), where the learner must balance the cost of incorrect decisions against the cost of continued testing. They show that the Bayes-optimal procedure for such problems is the sequential probability ratio test (SPRT) of Wald [46], with optimal thresholds determined by solving complex implicit equations that depend on the specific problem parameters. A number of works [44, 12, 5, 31] have extended this study to the more general composite hypothesis testing framework ($H_1 : \mu_1 - \mu_2 > 0$ vs. $H_2 : \mu_1 - \mu_2 \leq 0$). While there are similarities to our proposed setting, their analyses have been restricted to developing procedures that are only asymptotically Bayes-optimal and only hold in the case of exponential families.

**Paper organization.** The remainder of this paper is organized as follows. In §2, we study the problem in the 2-arm setting. This new formalism for BAI introduces novel intuitions which are best illustrated in the two arm setting. In §3, we present our algorithm and main results in the $K$-arm setting. Finally, in §4, we evaluate our methods on simulations and show that it outperforms traditional BAI methods on this problem.

## 2 Two-Arm Setting

To build intuition for this problem, we first study the $K = 2$ setting. Let $\mathcal{P}(\mathbb{R})$ denote all probability measures on $\mathbb{R}$, and let $G_\sigma = \left\{ \lambda \in \mathcal{P}(\mathbb{R}) : \forall t > 0, \ \mathbb{P}_\lambda \left( X - \mathbb{E}_\lambda[X] > t \right) \leq \exp\left( -t^2/2\sigma^2 \right) \right\}$ denote all $\sigma$-sub-Gaussian probability distributions. Let $\mathcal{M}$, defined below in (2), denote the class of two-armed bandit models with $\sigma$-sub-Gaussian rewards; recall that $\mu_i = \mathbb{E}_{\nu_i}[X]$. For a given gap $\Delta \geq 0$, let $\mathcal{M}_\Delta$, defined below, denote the subclass of models with sub-optimality gap $\Delta$. We have:

$$\mathcal{M} := \{\nu = (\nu_1, \nu_2) : \nu_1, \nu_2 \in G_\sigma; \mu_1, \mu_2 \in [0, B]\}, \quad \mathcal{M}_\Delta := \{\nu \in \mathcal{M} : \mu_1 - \mu_2 = \Delta\}. \quad (2)$$

In §2.1, we begin by studying $\mathcal{R}_{\mathrm{MI}}$ in (1), which uses the probability of misidentification as the performance criterion. In §2.2, we then consider $\mathcal{R}_{\mathrm{SR}}$, which instead uses the simple regret. Unless otherwise stated, all results in this section will be corollaries of more general results in §3.

### 2.1 Probability of misidentification in the two-arm setting

**Lower bound.** We begin with a gap-dependent lower bound applicable to any policy on this problem.

**Corollary 1.1** (Corollary of Theorem 1, Lower bound on $\mathcal{R}_{\mathrm{MI}}$)**.** *Fix a gap $\Delta > 0$ and the cost $c$ per arm pull. Then, for any policy $\pi$, we have*

$$\sup_{\nu \in \mathcal{M}_\Delta} \mathcal{R}_{\mathrm{MI}}(\pi, \nu) \geq \mathrm{LB}_{\mathrm{MI}}(\Delta) := \begin{cases} \frac{\sigma^2 c}{4\Delta^2} \log\left(\frac{e\Delta^2}{\sigma^2 c}\right), & \text{if } \Delta \geq \sqrt{\sigma^2 c}, \\ 1/4, & \text{if } \Delta < \sqrt{\sigma^2 c}. \end{cases} \quad (3)$$

It is instructive to compare the above result with lower bounds for fixed confidence BAI. As in the fixed confidence setting [30], we observe that for large $\Delta$, the lower bound exhibits a familiar

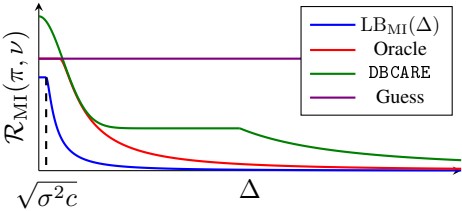 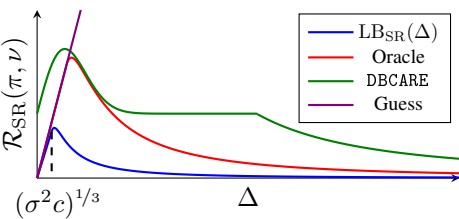

Figure 1: Illustrations of the lower and upper bounds on the risk for $\mathcal{R}_{\mathrm{MI}}$ (on the left) and $\mathcal{R}_{\mathrm{SR}}$ (on the right) in the 2-arm case presented throughout § 2, with the performance of the policy which guesses an arm at random without pulling at all (Guess) included as a point of reference.

dependence on $\sigma^2\Delta^{-2}$, indicating that the problem becomes easier as the gap increases. Our bound also depends on the cost $c$ and includes a logarithmic term in $\Delta^2(\sigma^2c)^{-1}$. Notably, when the gap is small, our setting departs from fixed confidence behavior: the lower bound undergoes a phase transition and saturates at a constant value of $1/4$, rather than continuing to increase with $\Delta^{-2}$.

**An oracular policy.** To build intuition towards designing a policy, it is worth considering the behavior of an "oracular" policy which knows the gap $\Delta$ but does not know which of the two arms is optimal. Recall that it requires approximately $N(\Delta, \delta) \in \mathcal{O}(\sigma^2\Delta^{-2}\log(1/\delta))$ samples to separate two sub-Gaussian distributions whose means are $\Delta$ apart [46, 22, 30] with probability at least $1 - \delta$. Hence, if we pull both arms $N(\Delta, \delta)$ times, we will incur a penalty of $\delta + \mathcal{O}(\sigma^2c\Delta^{-2}\log(1/\delta))$. By optimally choosing $\delta \in \mathcal{O}(\sigma^2c\Delta^{-2})$, we find that we need to pull each arm $\mathbb{N}(\Delta) \in \mathcal{O}(\sigma^2\Delta^{-2}\log(\Delta^2(\sigma^2c)^{-1}))$ times. However, the above expression is non-negative only when $\Delta \geq \Omega(\sqrt{\sigma^2c})$. Intuitively, if $\Delta$ is very small, the policy will need to incur a large cost to separate the two arms. If the policy knows $\Delta$ it is better off randomly guessing an arm instead of incurring this large cost. This intuition leads to the following policy and theoretical result. Its proof, which is straightforward, is given in Appendix C.

**Proposition 1.** *Let $\pi_\Delta$ be the policy which pulls each arm* $\max\left\{0, \left\lceil\frac{4\sigma^2}{\Delta^2}\log\left(\frac{\Delta^2}{8\sigma^2c}\right)\right\rceil\right\}$ *times. If it pulls $0$ times, it will choose an arm uniformly at random, and otherwise, outputs the empirically largest arm (breaking ties arbitrarily). Then, letting $\mathrm{LB}_{\mathrm{MI}}(\Delta)$ be as in (3), we have,*

$$\sup_{\nu\in\mathcal{M}_\Delta} \mathcal{R}_{\mathrm{MI}}(\pi_\Delta, \nu) \leq 32\mathrm{LB}_{\mathrm{MI}}(\Delta) + 2c \in \mathcal{O}(\mathrm{LB}_{\mathrm{MI}}(\Delta))$$

As we see, and illustrated in Fig 1, this bound matches the lower bound up to constant factors.[1] To design a policy when $\Delta$ is unknown, we will leverage the above intuition. We will also draw inspiration from prior work on racing-style algorithms [36, 34], which have shown that sequentially pulling arms and eliminating them based on confidence intervals can match oracular policies up to logarithmic factors in the fixed confidence setting.

**A policy for $\mathcal{R}_{\mathrm{MI}}$.** We will let $\delta$ be a confidence hyperparameter, aiming to output the optimal arm with probability at least $1 - \delta$. However, to avoid over-pulling when the gap $\Delta$ is too small, we also incorporate a hyperparameter $N^\star$, which is a limit on the total amount of times we are willing to pull *each* arm. Intuitively, we know the cost grows linearly in the number of pulls, but the probability of misidentification decays exponentially, so there is a point where the trade-off between the cost of pulling and the increased precision these pulls provide no longer favors continuing to pull.

Our approach proceeds in epochs of sampling both arms once and comparing the difference between the empirical averages of the two arms against a Hoeffding confidence bound at the end of each epoch to test for separation. If the observed difference on any epoch is larger than the confidence bound, it will exit and recommend the larger arm. Otherwise, it will continue to sample each arm until reaching the $N^\star$-th epoch, where it will return the arm with the larger empirical average even though they have not statistically separated. In the case of the 2-arm probability of misidentification setting, we use $N^\star = (2ec)^{-1}$ and $\delta = c(1 + 2cN^\star)^{-1}$. Here, we set $N^\star$ to be the maximum number of times the oracular policy would ever pull each arm for any $\Delta$. The confidence parameter $\delta$ is used to control

---

[1]Proposition 1 includes an additive penalty corresponding to the cost of two extra pulls, and a similar additive term appears in all upper bounds. This is unavoidable in general, as even as $\Delta \to \infty$, each arm must be pulled at least once to identify it. While this can be formally incorporated in the lower bound, we omit it for simplicity.

the penalty of the policy on the event that the policy's confidence interval for the gap does not contain the true gap. We have described this algorithm formally in the $K$-arm setting in Algorithm 1.

As the corollary below demonstrates, by careful choice of $N^\star$ and $\delta$, we show that we can match the lower bound in Corollary 2.1 up to $\log(1/c)$ factors, for all values of $\Delta$. Based on the relationship between algorithmic performance and lower bounds in the BAI literature, we conjecture that this logarithmic gap is largely unavoidable, and could at best be reduced to a log-log factor [28, 21, 30].

**Corollary 2.1** (Corollary of Theorem 2, DBCARE under $\mathcal{R}_{\mathrm{MI}}$). *Let $\pi$ be the policy described above using $N^* = (2ec)^{-1}$ and $\delta = c(1 + 2cN^*)^{-1}$. Then, letting $\mathrm{LB}_{\mathrm{MI}}(\Delta)$ be as in (3),*

$$\sup_{\nu \in \mathcal{M}_\Delta} \mathcal{R}_{\mathrm{MI}}(\pi, \nu) \leq 128 \log\left(\frac{e+1}{(ec)^2}\right) \mathrm{LB}_{\mathrm{MI}}(\Delta) + 3c \in \mathcal{O}\left(\log(c^{-1}) \mathrm{LB}_{\mathrm{MI}}(\Delta)\right).$$

This bound and its comparison to the lower bound are illustrated in Fig 1. As we can see in Fig 1, by our choice of $N^\star$, our policy actually performs within a constant factor of the lower bound for small $\Delta$, and the $\log(1/c)$ factor is incurred mostly in the "moderate" $\Delta$ regime. After the sharp transition at the midpoint of the plot in Fig 1, representing the point at which our algorithm is guaranteed to output the optimal arm before reaching $N^\star$ epochs with high probability, we can also see that the comparison to the lower bound quickly improves until we again reach a constant factor mismatch.

## 2.2 Simple regret in the two-arm setting

**Lower bound.** We again begin by presenting a lower bound on this problem.

**Corollary 3.1** (Corollary of Theorem 3, Lower bound on $\mathcal{R}_{\mathrm{SR}}$). *Fix a gap $\Delta > 0$ and the cost $c$ per arm pull. Then, for any policy $\pi$,*

$$\sup_{\nu \in \mathcal{M}_\Delta} \mathcal{R}_{\mathrm{SR}}(\pi, \nu) \geq \mathrm{LB}_{\mathrm{SR}}(\Delta) = \begin{cases} \frac{\sigma^2 c}{4\Delta^2} \log\left(\frac{e\Delta^3}{\sigma^2 c}\right), & \text{if } \Delta \geq (\sigma^2 c)^{1/3} \\ \Delta/4, & \text{if } \Delta < (\sigma^2 c)^{1/3} \end{cases} \tag{4}$$

*Additionally, taking the worst-case over all $\Delta$, we have, for any policy $\pi$,*

$$\sup_{\nu \in \mathcal{M}} \mathcal{R}_{\mathrm{SR}}(\pi, \nu) \geq \mathrm{LB}_{\mathrm{SR}}^\star = \frac{3}{8}\left(\frac{\sigma^2 c}{e}\right)^{1/3} \tag{5}$$

As in Corollary 1.1, we observe a phase transition in the lower bound: it is $\Delta/4$ when the gap is small, and scales as $\Omega(\Delta^{-2})$ when the gap is large. For what follows, we also state the minimax (worst-case) value of this lower bound as a function of $\Delta$. As we see, this minimax lower bound decreases as the arm-pull cost $c$ decreases. In contrast, for $\mathcal{R}_{\mathrm{MI}}$, the minimax lower bound is $1/4$, and even a naive policy that guesses an arm without any pulls incurs a penalty of only $1/2$. However, for $\mathcal{R}_{\mathrm{SR}}$, even achieving the minimax lower bound requires a well-designed policy.

**An oracular policy.** To design such a policy, let us again consider the behavior of an oracular policy which knows $\Delta$. The motivation behind the chosen number of samples is the same as before, but when pulling the arms $N(\Delta, \delta)$ times, we now incur a penalty of $\delta\Delta + \mathcal{O}(\sigma^2 c\Delta^{-2} \log(1/\delta))$. Because of this change, we now wish to use $\delta \in \mathcal{O}(\sigma^2 c\Delta^{-3})$, leading to the following result, mirroring that of Proposition 1. Its proof, which is straightforward, is given in Appendix C.

**Proposition 2.** *Let $\pi^\star$ be the policy which pulls each arm $\max\left\{0, \left\lceil \frac{4\sigma^2}{\Delta^2} \log\left(\frac{\Delta^3}{8\sigma^2 c}\right) \right\rceil\right\}$ times. If it pulls them 0 times, it will choose an arm uniformly at random, and otherwise, outputs the empirically largest arm (breaking ties arbitrarily). Then, letting $\mathrm{LB}_{\mathrm{SR}}(\Delta)$ be as in (4) and $\mathrm{LB}_{\mathrm{SR}}^\star$ as in (5),*

$$\sup_{\nu \in \mathcal{M}_\Delta} \mathcal{R}_{\mathrm{SR}}(\pi^\star, \nu) \leq 32 \mathrm{LB}_{\mathrm{SR}}(\Delta) + 2c \in \mathcal{O}(\mathrm{LB}_{\mathrm{SR}}(\Delta)), \quad \sup_{\nu \in \mathcal{M}} \mathrm{LB}_{\mathrm{SR}}(\pi^\star, \nu) \leq 8 \mathrm{LB}_{\mathrm{SR}}^\star + 2c$$

**A policy for $\mathcal{R}_{\mathrm{SR}}$.** Our policy will proceed exactly as before, performing rounds of equal sampling until either we reach a prespecified number of epochs or we are able to identify the optimal arm with high probability. Like the oracular policy, though, the change in risk requires updating our hyperparameters $N^\star$ and $\delta$ to ensure that our algorithm still performs well in this setting. We again motivate our choice of $N^\star$ via the behavior of the oracular policy, choosing $N^\star = (3/2e)(\sigma/c)^{2/3}$. We also still use $\delta$ as a tool to control the worst-case penalty when our confidence interval does not contain the true gap, and thus we set $\delta = c(B + 2cN^\star)^{-1}$.

**Corollary 4.1** (Corollary of Theorem 4, `DBCARE` under $\mathcal{R}_{\mathrm{SR}}$). *Let $\pi$ be the policy described above using $N^* = (3/2e)(\sigma/c)^{2/3}$ and $\delta = c(B + 2cN^\star)^{-1}$. Then, letting $\mathrm{LB}_{\mathrm{SR}}(\Delta)$ be as in (4), when $\Delta \geq (\sigma^2 c)^{1/3}$, we have,*

$$\sup_{\nu \in \mathcal{M}_\Delta} \mathcal{R}_{\mathrm{SR}}(\pi, \nu) \leq 128 \log\left(\frac{3B\sigma^{4/3}}{c^{5/3}}\right) \mathrm{LB}_{\mathrm{SR}}(\Delta) + 3c \in \mathcal{O}\left(\log(B\sigma c^{-1})\mathrm{LB}_{\mathrm{SR}}(\Delta)\right)$$

*When $\Delta < (\sigma^2 c)^{1/3}$, we instead have,*

$$\sup_{\nu \in \mathcal{M}_\Delta} \mathcal{R}_{\mathrm{SR}}(\pi, \nu) \leq 4\mathrm{LB}_{\mathrm{SR}}(\Delta) + 2(\sigma^2 c)^{1/3} + 3c \in \mathcal{O}\left(\mathrm{LB}_{\mathrm{SR}}(\Delta) + \mathrm{poly}(\sigma, c)\right)$$

*Finally, letting $\mathrm{LB}_{\mathrm{SR}}^\star$ be as in (5), taking the worst case over all $\Delta$, we have,*

$$\sup_{\nu \in \mathcal{M}} \mathcal{R}_{\mathrm{SR}}(\pi, \nu) \leq 9\mathrm{LB}_{\mathrm{SR}}^\star + 3c \in \mathcal{O}(\mathrm{LB}_{\mathrm{SR}}^\star)$$

Here we see, when $\Delta \geq (\sigma^2 c)^{1/3}$, these results closely mirror that of Corollary 2.1, though the log-factor now additionally scales with $B\sigma^2$. As illustrated in Fig 1, this log-factor primarily plays a role in the moderate $\Delta$ regime like in the case of $\mathcal{R}_{\mathrm{MI}}$. Our bound and Fig 1 also further highlight the inherent difficulty of designing a simultaneously minimax- and instance-optimal policy for $\mathcal{R}_{\mathrm{SR}}$, as it is impossible to match the lower bound as $\Delta \to 0$ without performing fewer pulls even as the problem becomes more difficult. Illustrating why the instance-based lower bound cannot be improved in this regime, however, is the policy which guesses an arm without any pulls in purple in Fig 1.

## 3 K-arm Setting

We now generalize our results to the $K$-arm setting. We begin by adapting the notation formalities for $K$ arms. We now let $\mathcal{M}$, defined in (6), denote the class of $K$-armed bandit models with $\sigma$-sub-Gaussian rewards. Further, for a bandit model $\nu \in \mathcal{M}$, assuming WLOG that we have $\mu_1 \geq \mu_2 \geq \cdots \geq \mu_K$, we define the complexity measure $\mathcal{H}(\nu) := \sum_{k=2}^{K} \Delta_k^{-2}$, where $\Delta_k = \mu_1 - \mu_k$ is the $k$-th largest suboptimality gap. For a given complexity $H > 0$, let $\mathcal{M}_H$, defined below, denote the subclass of models having complexity at most $H$. Thus, we define:

$$\mathcal{M} = \left\{\nu = (\nu_a)_{a=1}^K : \nu_a \in G_\sigma, \mu_a \in [0, B] \ \forall \ a \in [K]\right\}, \quad \mathcal{M}_H = \{\nu \in \mathcal{M} : \mathcal{H}(\nu) \leq H\} \quad (6)$$

As we will see, while our hardness results extend naturally from two to $K$ arms, extending the intuitions for the algorithm design requires a more careful design of the budget parameter $N^\star$.

### 3.1 Probability of misidentification in the K-arm setting

**Lower bound.** We now present the general $K$-arm lower bound result for $\mathcal{R}_{\mathrm{MI}}$.

**Theorem 1.** *Fix a complexity $H > 0$ and a cost per arm pull $c > 0$. Then, for any policy $\pi$,*

$$\sup_{\nu \in \mathcal{M}_H} \mathcal{R}_{\mathrm{MI}}(\pi, \nu) \geq \mathrm{LB}_{\mathrm{MI}}(H) = \begin{cases} \frac{\sigma^2 cH}{4} \log\left(\frac{e}{\sigma^2 cH}\right), & \text{if } H \leq (\sigma^2 c)^{-1} \\ 1/4, & \text{if } H > (\sigma^2 c)^{-1} \end{cases} \quad (7)$$

Comparing this result to its Corollary 1.1 in the 2-arm setting, we observe the same phase transition, now in terms of the complexity, $H$. Using the definition of $H$, we note that it still occurs when $\Delta_k \asymp \mathcal{O}((\sigma^2 c)^{-1})$, and it provides the same intuition: when at least some of the gaps are sufficiently close to zero (or if there are very many arms), the cost of separating them outweighs the decrease in the probability of misidentification, and it becomes optimal to guess the best arm without pulling.

**A policy for $\mathcal{R}_{\mathrm{MI}}$.** We present our proposed algorithm, `DBCARE`, in its full $K$-arm generality in Algorithm 1. To account for there now being $K$ arms, `DBCARE` maintains a "surviving set" $S$ of arms that have not yet been determined to be sub-optimal, and performs rounds of equal sampling of all arms in $S$. At the end of each round, it compares the difference between the current largest empirical average in $S$ and each other arm in $S$, and eliminates them based on Hoeffding confidence intervals. This continues until either there is only one arm remaining, or the remaining arms have reached their maximum per-arm budget, at which point the arm with the largest empirical average is returned.

In moving from the two arm to $K$-arm regimes, we once again encounter the issue of balancing performance and costs when selecting our per-arm budget. On one hand, if we naively replace the division by 2 in $N^\star$ in Corollary 2.1 with a division by $K$, then we will fall short on performance when there are many highly sub-optimal arms. However, if we keep the same budget for each arm from the 2-arm setting, we will perform too many total pulls when there are many near-optimal arms.

To this end, we allow the per-arm budgets to *adapt* to the problem complexity by letting $N^\star$ increase as $|S|$ decreases. This allows DBCARE to dedicate additional resources to separating the remaining arms as some are determined to be sub-optimal, but prevents the total possible number of pulls from scaling too quickly in $K$. Inspired by the 2-arm setting, we let $N^\star(k) = (kec)^{-1}$. Further, we still use the confidence $\delta$ to control the worst-case penalty when the confidence intervals do not contain the true gap, so we set $\delta = c(1 + 2c\log(K)N^\star(2))^{-1}$. The following theorem summarizes the key properties of DBCARE when applied to $\mathcal{R}_{\mathrm{MI}}$.

**Theorem 2.** *Let $\pi_{DBCARE}$ be the policy defined in Algorithm 1 using $N^\star(k) = (kec)^{-1}$ and $\delta = c(1 + 2c\log(K)N^\star(2))^{-1}$. Then, letting $\mathrm{LB}_{\mathrm{MI}}(H)$ be as in (7), we have,*

$$\sup_{\nu \in \mathcal{M}_H} \mathcal{R}_{\mathrm{MI}}(\pi_{DBCARE}, \nu) \leq 760 \log(K) \log\left(\frac{K\log(K)}{ec^2}\right) \mathrm{LB}_{\mathrm{MI}}(H) + (K+1)c,$$

*which is $\in \mathcal{O}(\mathrm{polylog}(K, c^{-1})\mathrm{LB}_{\mathrm{MI}}(H))$.*

As in the 2-arm case, we see in Theorem 2 that our policy is still able to achieve performance within a polylogarithmic factor of the lower bound, with the additional $\log(K)$ factor being due to the worst-case impact of our adaptive budget updating.

---

**Algorithm 1 D**ynamically **B**udgeted **C**ost-**A**dapted **R**isk-minimizing **E**limination

---

**Require:** Dynamic budget function $N^\star$, Confidence $\delta$
1: *Initialization*: $\hat{\mu}_k(0) = 0 \ \forall \ k \in [K], e_0 = 0, t = 0, n = 0, S = [K]$
2: **while** $n \leq N^\star(|S|)$ AND $|S| > 1$ **do**
3: $\quad n \leftarrow n + 1$
4: $\quad$ **for** $k \in S$ **do**
5: $\quad\quad t \leftarrow t + 1$
6: $\quad\quad A_t \leftarrow k$, Observe $X_t \sim \nu_{A_t}$
7: $\quad$ **end for**
8: $\quad \hat{\mu}_k(n) \leftarrow \frac{1}{n}\sum_{s=1}^{t} \mathbb{1}_{\{k\}}(A_s)X_s$, for $k \in S$
9: $\quad e_n \leftarrow \sqrt{4\sigma^2 n^{-1}\log(Kn\delta^{-1})}$.
10: $\quad S \leftarrow S \setminus \left\{ k \in S : \max_{\ell \in S} \hat{\mu}_\ell(n) - \hat{\mu}_k(n) > e_n \right\}$.
11: **end while**
12: **return** $\operatorname{argmax}_{a \in S} \hat{\mu}_a(n)$ (breaking ties randomly)

---

## 3.2 Simple regret in the K-arm setting

**Lower bound.** We now present our second lower bound, for $\mathcal{R}_{\mathrm{SR}}$ in the general $K$-arm setting.

**Theorem 3.** *Fix a complexity $H > 0$, a smallest gap $\Delta_2 \geq 0$, and a cost per arm pull $c > 0$. Then, for any policy $\pi$, we have,*

$$\sup_{\nu \in \mathcal{M}_H} \mathcal{R}_{\mathrm{SR}}(\pi, \nu) \geq \mathrm{LB}_{\mathrm{SR}}(H) = \begin{cases} \frac{c\sigma^2 H}{4}\log\left(\frac{e\Delta_2}{\sigma^2 cH}\right), & \text{if } H\Delta_2^{-1} \leq (\sigma^2 c)^{-1} \\ \Delta_2/4, & \text{if } H\Delta_2^{-1} > (\sigma^2 c)^{-1} \end{cases} \tag{8}$$

*Additionally, taking the worst case over all problem instances, we have, for any policy $\pi$,*

$$\sup_{\nu \in \mathcal{M}} \mathcal{R}_{\mathrm{SR}}(\pi, \nu) \geq \mathrm{LB}_{\mathrm{SR}}^\star = \frac{3}{8}\left(\frac{(K-1)\sigma^2 c}{e}\right)^{1/3} \tag{9}$$

Looking at the bound presented in (8), we see that the phase transition in this lower bound now jointly involves the total problem complexity and the smallest gap.

**A policy for $\mathcal{R}_{\mathrm{SR}}$.** Following the same intuition as in the probability of misidentification case, we again wish to allow $N^\star$ to adapt to the problem complexity and increase as the surviving set of arms shrinks. Observing the minimax lower bound presented in 9, though, we see that the maximum problem difficulty scales with $K^{1/3}$, unlike the constant scaling in the case of $\mathcal{R}_{\mathrm{MI}}$. Thus, we wish for $N^\star$ to scale with $K^{-2/3}$ instead of $K^{-1}$, and so we choose $N^\star(k) = (3/2e)\sigma^{2/3}((k-1)c)^{-2/3}$. Then, controlling for the worst-case performance again, we choose $\delta = c(B + eK^{1/3}\log(K)N^\star(2))^{-1}$.

**Theorem 4.** *Let $\pi_{DBCARE}$ be the policy defined in Algorithm 1 using $N^\star(k) = (3/2e)\sigma^{2/3}((k-1)c)^{-2/3}$ and $\delta = c(B + eK^{1/3}\log(K)N^\star(2))^{-1}$. Then, letting $\mathrm{LB}_{\mathrm{SR}}$ be as in (8), when $H\Delta_2^{-1} \leq (\sigma^2 c)^{-1}$,*

$$\sup_{\nu \in \mathcal{M}_H} \mathcal{R}_{\mathrm{SR}}(\pi_{DBCARE}, \nu) \leq 550 \log(K) \log\left(\frac{K\log(K)B\sigma^{4/3}}{c^{5/3}}\right) \mathrm{LB}_{\mathrm{SR}}(H) + (K+1)c,$$

*which is $\in \mathcal{O}(\mathrm{polylog}(B, K, \sigma, c^{-1})\mathrm{LB}_{\mathrm{SR}}(H))$. When $H\Delta_2^{-1} > (\sigma^2 c)^{-1}$, we instead have,*

$$\mathrm{LB}_{\mathrm{SR}}(H) + 4\log(K)(K\sigma^2 c)^{1/3} + (K+1)c \in \mathcal{O}(\mathrm{LB}_{\mathrm{SR}}(H) + \log(K)\mathrm{poly}(K, \sigma, c))$$

*Finally, letting $\mathrm{LB}_{\mathrm{SR}}^\star$ be as in (9), we have,*

$$\sup_{\nu \in \mathcal{M}} \mathcal{R}_{\mathrm{SR}}(\pi_{DBCARE}, \nu) \leq 20\log(K)\mathrm{LB}_{\mathrm{SR}}^\star + (K+1)c \in \mathcal{O}(\log(K)\mathrm{LB}_{\mathrm{SR}}^\star)$$

In Theorem 4, we see similar performance of DBCARE compared to the lower bound as in the two-arm case: we are able to achieve performance within polylogarithmic factors when the complexity is relatively low, and we incur an additive logarithmic and polynomial factor in $K, \sigma$, and $c$ when the complexity is prohibitively high. Observing our comparison to the minimax bound, we see that our policy is still minimax-optimal, being only a logarithmic factor in $K$ beyond the lower bound.

## 4 Numerical Experiments

**Simulation Studies.** We now empirically compare our method against traditional fixed budget and fixed confidence methods to demonstrate the ability of DBCARE to perform well across all problem instances. We study the performance across a range of suboptimality gaps $\Delta$ for Gaussian and Bernoulli rewards in the two-arm setting using the cost $c = 10^{-4}$. In the Gaussian setting, the arms have variance $\sigma^2 = 1$ with means $\pm\Delta/2$, for $\Delta \in [0.05, 2]$; for Bernoulli arms, the means are $0.5\pm\Delta/2$, for $\Delta \in [0.01, 0.95]$. Results are averaged across $10^5$ runs each with different random seeds. We compare to Sequential Halving for fixed budget and elimination procedures using the optimized stopping rules of [30] for fixed confidence. We use budgets $T = 10$ and $T = 500$ and confidences of $\delta = 0.1$ and $\delta = 0.01$ for comparison against relatively low and high confidence/budget choices. We also include the oracular policies of § 2 to provide a baseline of good performance. As we can see in Fig 2, the fixed budget and confidence algorithms necessarily have some region of gaps where they perform sub-optimally: for the small budget, it is moderate $\Delta$ values, for the large budget, it is large $\Delta$ values, and for both confidences, it is small $\Delta$ values. On the contrary, our proposed algorithm exhibits uniformly good performance across all $\Delta$ values, which is preferable when $\Delta$ is unknown. In Appendix E, we provide further simulations in the $K$-arm setting.

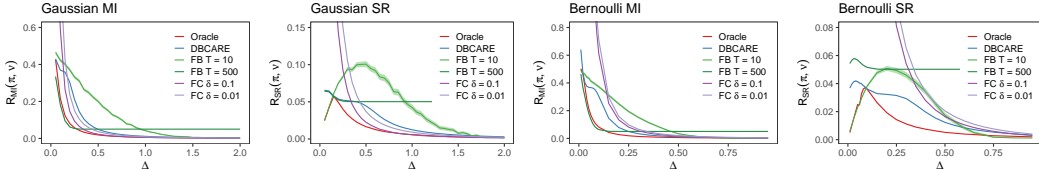

Figure 2: Comparisons between the oracular policy, DBCARE, and fixed budget and confidence algorithms for $\mathcal{R}_{\mathrm{MI}}$ and $\mathcal{R}_{\mathrm{SR}}$. $Y$-axes are adjusted per setting to highlight problem-specific behavior. Confidence regions represent empirical average risk $\pm$ 2 SE.

**Drug Discovery Experiment.** To demonstrate the efficacy of our approach on a problem in practice, we present the results of a real data experiment on a drug discovery dataset. For this experiment, we take the results from Table 2 of Genovese et al. [19] on the efficacy of the drug secukinumab in patients

with rheumatoid arthritis. They report outcomes for 237 patients assigned to one of 5 treatment groups (arms) and report the drug efficacy according to the American College of Rheumatology criteria ACR20, ACR50, and ACR70. We consider this data under 2 settings: 1.) a binary efficacy outcome, being whether a patient achieved at least ACR20 (1) or not (0), as this was the primary goal of [19]; and 2.) a "leveled" efficacy outcome, where no improvement results in an outcome of 0, and ACR20, ACR50, and ACR70 are outcomes of 0.2, 0.5, and 0.7, respectively, approximating a continuous efficacy metric. We treat the proportions of patients reported in each category in Table 2 of [19] as population proportions, and evaluate DBCARE, Sequential Halving, and an elimination procedure with confidence bounds $\sqrt{4\sigma^2 n^{-1} \log(Kn\delta^{-1})}$ on $10^4$ runs in each setting, each with different random seeds. For the binary outcome setting, the means were $\mu = (0.537, 0.469, 0.465, 0.360, 0.340)$, and for the leveled outcome setting, the means were $\mu = (0.230, 0.227, 0.200, 0.196, 0.102)$, each presented in decreasing order (order was randomized during data generation). Because $K$ and the means are fixed in this setting, we choose to evaluate our performance across a range of values for $c \in [10^{-3}, 10^{-5}]$. In Fig 3, we can see that no other method uniformly outperforms DBCARE across all choices of $c$, again highlighting the ability of our method to adapt to the problem setting at hand.

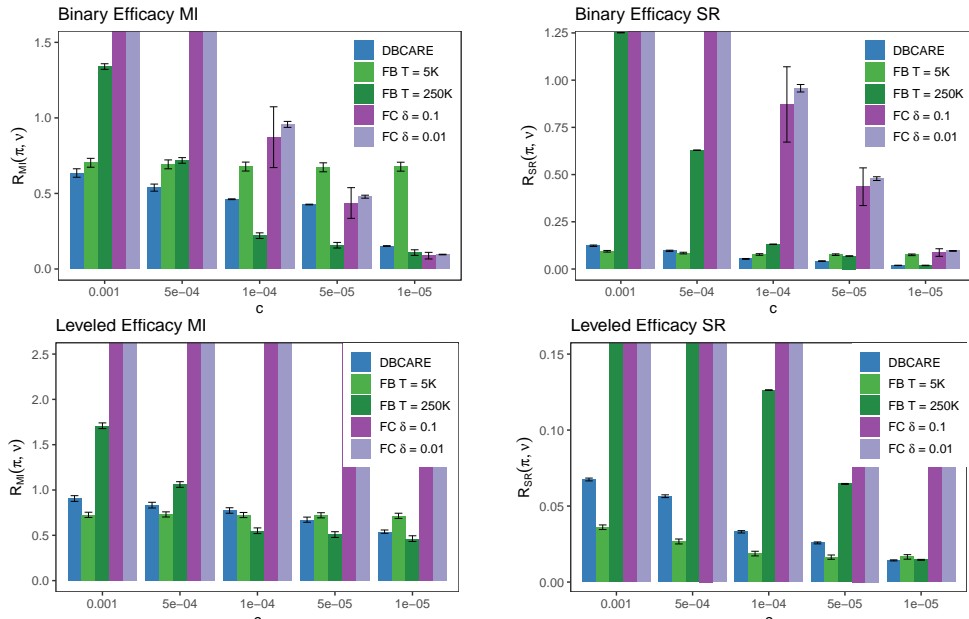

Figure 3: Comparisons between DBCARE and fixed budget and confidence algorithms for $\mathcal{R}_{\mathrm{MI}}$ and $\mathcal{R}_{\mathrm{SR}}$ on a drug discovery dataset. $Y$-axes are adjusted per setting to highlight problem-specific behavior. Error bars represent empirical average risk $\pm$ 2 SE.

## 5    Conclusion

We propose a novel framework for studying best arm identification. In many practical settings, the traditional fixed budget and confidence regimes do not nicely align with the objectives of practitioners. To fill this gap, our setting explicitly balances sampling costs and performance on the fly by minimizing a risk functional. We prove hardness results for this problem and provide an algorithm, DBCARE, which achieves near-optimal performance on nearly all problem instances.

**Future directions.** We believe our lower bound analysis for simple regret in the $K$-arm setting can be improved. Though our bounds are tight when suboptimality gaps are similar, we believe the bounds can be tighter when they are different. We also conjecture that the additive gap we observe in the simple regret setting is unavoidable for algorithms which achieve the minimax risk.

## Acknowledgments and Disclosure of Funding

This work was supported in part by NSF Awards IIS-2441796 and DMS-2023239.

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

## A  Results from Prior Works

**Lemma 1** (Lemma 18 of [30]). *Let $\nu$ and $\nu'$ be two bandit models, and let $\tau$ be any stopping time with respect to $\mathcal{F}_t$, where $\mathcal{F}_t = \sigma(A_1, X_1, \ldots, A_t, X_t)$ is the sigma-algebra generated by all bandit interactions. For every event $\mathcal{E} \in \mathcal{F}_\tau$ (i.e., $\mathcal{E}$ such that $\mathcal{E} \cap \{\tau = t\} \in \mathcal{F}_t$),*

$$\mathbb{P}_{\nu'}(\mathcal{E}) = \mathbb{E}_\nu[\mathbb{1}_\mathcal{E} \exp(-L_\tau)],$$

*where,*

$$L_\tau = \sum_{a=1}^{K} \sum_{s=1}^{N_a(\tau)} \log\left(\frac{f_a(Y_{a,s})}{f_a'(Y_{a,s})}\right),$$

*where $Y_{a,s}$ is the $s$-th i.i.d. observation of the $a$-th arm and $f_a$ and $f_a'$ are the distribution functions of the $a$-th arm under $\nu$ and $\nu'$, respectively.*

**Lemma 2** (Lemma 4 of [9]). *Let $\rho_0, \rho_1$ be two probability distributions supported on some set $\mathcal{X}$, with $\rho_1 \ll \rho_0$. Then, for any measurable function $\phi : \mathcal{X} \to \{0, 1\}$, one has*

$$\mathbb{P}_{X \sim \rho_0}(\phi(X) = 1) + \mathbb{P}_{X \sim \rho_1}(\phi(X) = 0) \geq \frac{1}{2}\exp(-\mathrm{KL}(\rho_0 \,||\, \rho_1)).$$

## B  Proof of Lower Bounds

We begin with a Lemma that will be central to all of our lower bounds, which builds upon the work of [30] to achieve a bound with the form of their Lemma 15 which admits a random stopping time.

**Lemma 3.** *Let $\nu$ and $\nu'$ be two $K$-arm bandit models. Let $\pi$ be any policy with associated stopping time $\tau$ such that $\mathbb{P}(\tau < \infty) = 1$ which outputs an arm $\widehat{I} \in [K]$ at time $\tau$. Then for any $a \in [K]$,*

$$\mathbb{P}_{\nu,\pi}(\widehat{I} \neq a) + \mathbb{P}_{\nu',\pi}(\widehat{I} = a) \geq \frac{1}{2}\exp\left(-\sum_{a=1}^{K} \mathbb{E}_{\nu,\pi}[N_a(\tau)]\,\mathrm{KL}(\nu_a \,||\, \nu_a')\right),$$

*where $\mathbb{P}_{\nu,\pi}$ is the probability with respect to all randomness incurred by $\pi$ interacting with the bandit model $\nu$ and $N_a(t) = \sum_{s=1}^{t} \mathbb{1}_{\{a\}}(A_s)$ is the number of times arm $a$ has been pulled up to round $t$.*

*Proof.* Fix a policy $\pi$. For ease of notation, we suppress the $\pi$ in the probabilities and expectations throughout the proof. Now, we begin by using Lemma 1 to prove that the distributions of $\widehat{I}$ under each bandit model are absolutely continuous with one another. Consider an event $\mathcal{E} \in \mathcal{F}_\tau$ such that $\mathbb{P}_{\nu'}(\mathcal{E}) = 0$. Then, because $e^{-x} > 0$ for all $x \in \mathbb{R}$, we must have $\mathbb{P}_\nu(\mathbb{1}_\mathcal{E} \exp(-L_\tau) = 0) = 1$ by Lemma 1. Further, by our supposition that $\mathbb{P}_\nu(\tau < \infty) = 1$, we must have $\mathbb{P}_\nu(\exp(-L_\tau) > 0) = 1$, and so we have $\mathbb{P}_{\nu'}(\mathcal{E}) = 0 \implies \mathbb{P}_\nu(\mathcal{E}) = 0$, and we achieve the reverse implication by symmetry. Now, consider the fact that we necessarily have $\{\widehat{I} = a\} \in \mathcal{F}_\tau$ for all $a \in [K]$ by construction, and so if we denote by $\mathcal{L}(\widehat{I})$ and $\mathcal{L}'(\widehat{I})$ the distributions of $\widehat{I}$ under $\nu$ and $\nu'$, respectively, then clearly $\mathcal{L}'(\widehat{I}) \ll \mathcal{L}(\widehat{I})$. Thus, we can apply Lemma 2 to show,

$$\mathbb{P}_\nu(\widehat{I} \neq a) + \mathbb{P}_{\nu'}(\widehat{I} = a) \geq \frac{1}{2}\exp\left(-\mathrm{KL}(\mathcal{L}(\widehat{I}) \,||\, \mathcal{L}'(\widehat{I}))\right)$$

To conclude the proof, we need only upper bound $\mathrm{KL}(\mathcal{L}(\widehat{I}) \,||\, \mathcal{L}'(\widehat{I}))$ by $\mathbb{E}_\nu[L_\tau]$, which we know is equal to $\sum_{a=1}^{K} \mathbb{E}_\nu[N_a(\tau)]\,\mathrm{KL}(\nu_a \,||\, \nu_a')$ by an application of Wald's Lemma [42]. By applying the conditional Jensen inequality to the statement of Lemma 1 and rearranging the terms, we know for any $\mathcal{E} \in \mathcal{F}_\tau$, we have $\mathbb{E}_\nu[L_\tau \mid \mathcal{E}] \geq \log\frac{\mathbb{P}_\nu(\mathcal{E})}{\mathbb{P}_{\nu'}(\mathcal{E})}$. Thus, letting $\mathcal{I} = \{k \in [K] : \mathbb{P}_\nu(\widehat{I} = k) \neq 0\}$, we can write,

$$\begin{aligned}
\mathbb{E}_\nu[L_\tau] &= \sum_{k \in \mathcal{I}} \mathbb{E}_\nu[L_\tau \mid \widehat{I} = k]\,\mathbb{P}_\nu(\widehat{I} = k) \\
&\geq \sum_{k \in \mathcal{I}} \log\left(\frac{\mathbb{P}_\nu(\widehat{I} = k)}{\mathbb{P}_{\nu'=k}(\widehat{I} = k)}\right)\mathbb{P}_\nu(\widehat{I} = k) \\
&= \mathrm{KL}(\mathcal{L}(\widehat{I}) \,||\, \mathcal{L}'(\widehat{I})),
\end{aligned}$$

which concludes the proof. $\qquad\square$

We now employ Lemma 3 to prove Theorems 1 and 3 and their associated corollaries.

*Proof of Theorem 1.* Fix $H > 0, \sigma^2 > 0, c > 0$, and a policy $\pi$. Let $\nu$ be a Gaussian $K$-arm bandit model with means $\mu_1 > \mu_2 \geq \cdots \geq \mu_K$ and common variance $\sigma^2$ satisfying $\mathcal{H}(\nu) = H$. Then, it is easy to show by contradiction that there must exist some arm $a \in \{2, \ldots, K\}$ such that $\mathbb{E}_{\pi,\nu}[N_a(\tau)] \leq \frac{\mathbb{E}_{\pi,\nu}[\tau]}{\Delta_a^2 H(\nu)}$. Let $\nu'$ be an alternative model with Gaussian arms having the same common variance $\sigma^2$, where $\mu_k(\nu) = \mu_k(\nu')$ for all $k \neq a$ and $\mu_a(\nu') = \mu_a(\nu) + 2\Delta_a$ so that arm $a$ is now the optimal arm. Clearly $\mathcal{H}(\nu') \leq \mathcal{H}(\nu)$, and so we have $\nu, \nu' \in \mathcal{M}_H$. Then, we can apply Lemma 3 to show,

$$\sup_{\nu \in \mathcal{M}_H} \mathcal{R}_{\mathrm{MI}}(\pi, \nu) \geq \max\{\mathcal{R}_{\mathrm{MI}}(\pi, \nu), \mathcal{R}_{\mathrm{MI}}(\pi, \nu')\}$$

$$\geq \frac{1}{2}\left(\mathcal{R}_{\mathrm{MI}}(\pi, \nu) + \mathcal{R}_{\mathrm{MI}}(\pi, \nu')\right)$$

$$= \frac{1}{2}\left(\mathbb{P}_{\nu,\pi}(\widehat{I} \neq 1) + \mathbb{P}_{\nu',\pi}(\widehat{I} \neq a)\right) + \frac{c}{2}\left(\mathbb{E}_{\nu,\pi}[\tau] + \mathbb{E}_{\nu',\pi}[\tau]\right)$$

$$\geq \frac{1}{2}\left(\mathbb{P}_{\nu,\pi}(\widehat{I} \neq 1) + \mathbb{P}_{\nu',\pi}(\widehat{I} = 1)\right) + \frac{c}{2}\left(\mathbb{E}_{\nu,\pi}[\tau] + \mathbb{E}_{\nu',\pi}[\tau]\right)$$

$$\geq \frac{1}{4}\exp\left(-\sum_{k=1}^K \mathbb{E}_{\nu,\pi}[N_k(\tau)]\,\mathrm{KL}(\nu_k \,||\, \nu_k')\right) + \frac{c}{2}\left(\mathbb{E}_{\nu,\pi}[\tau] + \mathbb{E}_{\nu',\pi}[\tau]\right)$$

$$\geq \frac{1}{4}\exp\left(-\frac{2\,\mathbb{E}_{\nu,\pi}[\tau]}{\sigma^2 H}\right) + \frac{c}{2}\left(\mathbb{E}_{\nu,\pi}[\tau] + \mathbb{E}_{\nu',\pi}[\tau]\right) \tag{10}$$

Here, we recognize the fact that (10) is convex in $\mathbb{E}_{\nu,\pi}[\tau]$, and thus we provide a $\pi$-free lower bound by minimizing over $\mathbb{E}_{\nu,\pi}[\tau], \mathbb{E}_{\nu',\pi}[\tau] \geq 0$, which is achieved by $\mathbb{E}_{\nu,\pi}[\tau] = \max\{0, \frac{\sigma^2 H}{2}\log(1/\sigma^2 cH)\}$ and $\mathbb{E}_{\nu',\pi}[\tau] = 0$. Plugging in these values completes the proof. $\square$

*Proof of Theorem 3.* This proof proceeds nearly identically to the proof above. Again fix $H > 0, \sigma^2 > 0, c > 0$ and any policy $\pi$. Then, let $\nu$ and $\nu'$ be the same Gaussian $K$-arm bandit models as in the previous proof, with optimal arms 1 and $a \in \{2, \ldots, K\}$, respectively. For notational clarity, let the suboptimality gaps $\Delta_2, \ldots, \Delta_K$ be with respect to $\nu$ and let $\Delta_1 \equiv 0$, so that $\mu_a(\nu') - \mu_k(\nu') = \Delta_a + \Delta_k$ for $k \neq a$. Then, again using Lemma 3, we have,

$$\sup_{\nu \in \mathcal{M}_H} \mathcal{R}_{\mathrm{SR}}(\pi, \nu) \geq \max\{\mathcal{R}_{\mathrm{SR}}(\pi, \nu), \mathcal{R}_{\mathrm{SR}}(\pi, \nu')\}$$

$$\geq \frac{1}{2}\left(\mathcal{R}_{\mathrm{SR}}(\pi, \nu), \mathcal{R}_{\mathrm{SR}}(\pi, \nu')\right)$$

$$= \frac{1}{2}\left(\mathbb{E}_{\nu,\pi}[\mu_1 - \mu_{\widehat{I}}] + \mathbb{E}_{\nu',\pi}[\mu_a - \mu_{\widehat{I}}]\right) + \frac{c}{2}\left(\mathbb{E}_{\nu,\pi}[\tau] + \mathbb{E}_{\nu',\pi}[\tau]\right)$$

$$= \frac{1}{2}\left(\sum_{i=2}^K \Delta_i\,\mathbb{P}_{\nu,\pi}(\widehat{I} = i) + \sum_{j \neq a}(\Delta_a + \Delta_j)\,\mathbb{P}_{\nu',\pi}(\widehat{I} = j)\right)$$

$$\quad + \frac{c}{2}\left(\mathbb{E}_{\nu,\pi}[\tau] + \mathbb{E}_{\nu',\pi}[\tau]\right)$$

$$\geq \frac{\Delta_2}{2}\left(\mathbb{P}_{\nu,\pi}(\widehat{I} \neq 1) + \mathbb{P}_{\nu',\pi}(\widehat{I} \neq a)\right) + \frac{c}{2}\left(\mathbb{E}_{\nu,\pi}[\tau] + \mathbb{E}_{\nu',\pi}[\tau]\right)$$

$$\geq \frac{\Delta_2}{2}\left(\mathbb{P}_{\nu,\pi}(\widehat{I} \neq 1) + \mathbb{P}_{\nu',\pi}(\widehat{I} = 1)\right) + \frac{c}{2}\left(\mathbb{E}_{\nu,\pi}[\tau] + \mathbb{E}_{\nu',\pi}[\tau]\right)$$

$$\geq \frac{\Delta_2}{4}\exp\left(-\sum_{k=1}^K \mathbb{E}_{\nu,\pi}[N_k(\tau)]\,\mathrm{KL}(\nu_k \,||\, \nu_k')\right) + \frac{c}{2}\left(\mathbb{E}_{\nu,\pi}[\tau] + \mathbb{E}_{\nu',\pi}[\tau]\right)$$

$$\geq \frac{\Delta_2}{4}\exp\left(-\frac{2\,\mathbb{E}_{\nu,\pi}[\tau]}{\sigma^2 H}\right) + \frac{c}{2}\left(\mathbb{E}_{\nu,\pi}[\tau] + \mathbb{E}_{\nu',\pi}[\tau]\right) \tag{11}$$

Once again, (11) is convex in $\mathbb{E}_{\nu,\pi}[\tau]$, so we can further lower bound (11) by setting $\mathbb{E}_{\nu,\pi}[\tau] = \max\{0, \frac{\sigma^2 H}{2}\log(\Delta_2(\sigma^2 cH)^{-1})\}$ and $\mathbb{E}_{\nu',\pi}[\tau] = 0$, completing the proof of (8).

To prove (9), we can consider a specific set of means satisfying this problem instance. Consider the instance where $\mu_1 = \Delta$ and $\mu_2 = \cdots = \mu_K = 0$, so that $\Delta_2 = \cdots = \Delta_K = \Delta$ for some $\Delta > 0$ that we will specify later. With these means, we have $H = (K-1)\Delta^{-2}$. Then, using (8), we can write,

$$\sup_{\nu \in \mathcal{M}_{(K-1)\Delta^{-2}}} \mathcal{R}_{\mathrm{SR}}(\pi, \nu) \geq \mathrm{LB}_{\mathrm{SR}}((K-1)\Delta^{-2})$$

$$= \begin{cases} \frac{(K-1)\sigma^2 c}{4\Delta^2} \log\left(\frac{e\Delta^3}{(K-1)\sigma^2 c}\right), & \text{if } \Delta \geq ((K-1)\sigma^2 c)^{1/3} \\ \Delta/4, & \text{if } \Delta < ((K-1)\sigma^2 c)^{1/3} \end{cases}$$

We can then find the $\Delta$ which maximizes this function, which occurs at $\Delta^\star = (\sqrt{e}(K-1)\sigma^2 c)^{1/3}$, which gives,

$$\sup_{\nu \in \mathcal{M}} \mathcal{R}_{\mathrm{SR}}(\pi, \nu) \geq \sup_{\nu \in \mathcal{M}_{(K-1)(\Delta^\star)^{-2}}} \mathcal{R}_{\mathrm{SR}}(\pi, \nu) \geq \mathrm{LB}_{\mathrm{SR}}((K-1)(\Delta^\star)^{-2}) = \frac{3}{8}\left(\frac{(K-1)\sigma^2 c}{e}\right)^{1/3}$$

$\square$

*Proof of Corollaries 1.1 and 3.1.* Recall that, when $K = 2$, $H = \Delta^{-2}$ and $\Delta_2 = \Delta$. The conclusions then follow directly from Theorems 1 and 3. $\square$

# C   Oracular Policy Proofs

*Proof of Proposition 1.* Fix a gap $\Delta > 0$. Because samples from each arm are i.i.d. $\sigma$-sub-Gaussian, by equally sampling the arms, we have i.i.d. $\sqrt{2}\sigma$-sub-Gaussian observations of the gap $\Delta$. By a Hoeffding confidence bound, if $\pi_T$ pulls each arm a fixed number of times $\lceil T \rceil$ and outputs the empirically largest arm, then we have

$$\mathcal{R}_{\mathrm{MI}}(\pi_T, \nu) = \mathbb{P}(\hat{\Delta}_{\lceil T \rceil} < 0) + 2c\lceil T \rceil \leq \exp\left(-\frac{T\Delta^2}{4\sigma^2}\right) + 2c(T+1)$$

Plugging in the proposed number of pulls, we get $\mathcal{R}_{\mathrm{MI}}(\pi_\Delta, \nu) \leq \frac{8\sigma^2 c}{\Delta^2}\log\left(\frac{e\Delta^2}{8\sigma^2 c}\right) + 2c$ when $\Delta \geq \sqrt{8\sigma^2 c}$, and exactly $\mathcal{R}_{\mathrm{MI}}(\pi_\Delta, \nu) = 1/2$ otherwise, as then the policy guesses the optimal arm uniformly at random. Multiplying (3) by 32 and adding $2c$ then clearly upper bounds this quantity. $\square$

*Proof of Proposition 2.* This proof proceeds nearly identically to the previous. Again fix a gap $\Delta > 0$, and consider that we can write,

$$\mathcal{R}_{\mathrm{SR}}(\pi_T, \nu) = \Delta\,\mathbb{P}(\hat{\Delta}_{\lceil T \rceil} < 0) + 2c\lceil T \rceil \leq \Delta\exp\left(-\frac{T\Delta^2}{4\sigma^2}\right) + 2c(T+1)$$

Then, plugging in the proposed number of pulls, we get $\mathcal{R}_{\mathrm{SR}}(\pi^\star, \nu) \leq \frac{8\sigma^2 c}{\Delta^2}\log\left(\frac{e\Delta^3}{8\sigma^2 c}\right) + 2c$ when $\Delta \geq (8\sigma^2 c)^{1/3}$ and exactly $\mathcal{R}_{\mathrm{SR}}(\pi^\star, \nu) = \Delta/2$ otherwise, as then the policy guesses the optimal arm uniformly at random. Multiplying (4) by 32 and adding $2c$ then clearly upper bounds this quantity. Further, maximizing this upper bound in terms of $\Delta$ (occurring at $\Delta = (8\sqrt{e}\sigma^2 c)^{1/3}$) yields,

$$\sup_{\nu \in \mathcal{M}} \mathcal{R}_{\mathrm{SR}}(\pi^\star, \nu) = \sup_{\Delta}\sup_{\nu \in \mathcal{M}_\Delta} \mathcal{R}_{\mathrm{SR}}(\pi^\star, \nu) \leq 3\left(\frac{\sigma^2 c}{e}\right)^{1/3} + 2c = 8\mathrm{LB}_{\mathrm{SR}}^\star + 2c$$

$\square$

# D   Upper Bounds for DBCARE

We begin by presenting a number of technical lemmas allowing us to control the behavior of DBCARE and prove our desired upper bounds on its performance.

**Lemma 4** (Bound on total number of pulls). *For any bandit instance $\nu$, using $N^\star(k) = (kec)^{-1}$ and $N^\star(k) = (3/2e)\sigma^{2/3}((k-1)c)^{-2/3}$, DBCARE satisfies,*

$$\mathbb{E}_{\nu,\pi}[\tau] \leq \frac{2\log(K)}{ec}, \qquad \mathbb{E}_{\nu,\pi}[\tau] \leq \frac{3\log(K)(K\sigma^2)^{1/3}}{2c^{2/3}},$$

*respectively.*

*Proof.* Let $\widehat{k}$ denote the index of the $k$-th arm eliminated by the algorithm. Then by construction, $\mathbb{E}_{\nu,\pi}[N_{\widehat{k}}(\tau)] \leq N^\star(K - k + 1)$. Further, $\mathbb{E}_{\nu,\pi}[N_{\widehat{I}}(\tau)] \leq N^\star(2)$. Thus,

$$\mathbb{E}_{\nu,\pi}[\tau] = \sum_{a=1}^{K} \mathbb{E}_{\nu,\pi}[N_a(\tau)] \leq N^\star(2) + \sum_{k=2}^{K} N^\star(k)$$

Then, apply the fact that $1/2 + \sum_{k=2}^{K} k^{-1} \leq 2\log(K)$ and $1 + \sum_{k=2}^{K}(k-1)^{-2/3} \leq eK^{1/3}\log(K)$ to prove the statements. $\qquad\square$

**Lemma 5** (Elimination behavior). *Consider a bandit instance $\nu$ satisfying, WLOG, $\mu_1 \geq \mu_2 \geq \cdots \geq \mu_K$. Let $n(t)$ be the epoch associated with time $t$. Define the good event,*

$$G = \bigcap_{n(t) \leq n(\tau)} \bigcap_{k \in S\backslash\{1\}} \{\Delta_k \in (\hat{\mu}_1(n(t)) - \hat{\mu}_k(n(t)) - e_{n(t)}, \hat{\mu}_1(n(t)) - \hat{\mu}_k(n(t)) + e_{n(t)})\}.$$

*Then,*

1. $\mathbb{P}_{\nu,\pi}(G^c) \leq \delta$

2. *On $G$, $1 \in S \ \forall \ n(t) \leq n(\tau)$ (i.e. the optimal arm is never eliminated)*

3. *On $G$, if $\Delta_k > \sqrt{\frac{16\sigma^2 \log(KN^\star(k)/\delta)}{N^\star(k)}}$ for all $k \geq \ell \in \{2, \ldots, K\}$ and $N^\star$ decreasing in $k$,*

$$N_k(\tau) \leq \frac{16\sigma^2 \log(KN^\star(k)/\delta)}{\Delta_k^2} < N^\star(k) \ \forall \ k \geq \ell$$

*Proof.*
**Part 1.** Letting $Y_{a,s}$ denote the $s$-th i.i.d. observation from arm $a$, by assumption, $Y_{1,s} - Y_{k,s}$ are $\sqrt{2}\sigma$-sub-Gaussian random variables with mean $\Delta_k$. Thus, $\sqrt{\frac{4\sigma^2 \log(n/\delta)}{n}}$ is a $\delta$-correct confidence interval width for $\Delta_k$ after $n$ observations using $\widehat{\Delta}_{k,n} = \hat{\mu}_1(n) - \hat{\mu}_k(n)$ as the point estimate [21, 30]. Replacing $\delta$ by $\delta/K$ and taking a union bound across all $k \in S \setminus \{1\}$ then proves 1. **Part 2.** Consider that on $G$, $\hat{\mu}_k(n) - \hat{\mu}_1(n) \leq e_n - \Delta_k \leq e_n$ for all $k \neq 1$, which proves 2. **Part 3.** We begin with arm $K$. By the supposition, on $G$, there exists $n < N^\star(k)$ such that $\hat{\mu}_1(n) - \hat{\mu}_K(n) - e_n \geq \Delta_K - 2e_n > 0$, and thus $K \notin S$ for all $m > n$. Further, we can upper bound the $n$ at which this is true by $\frac{16\sigma^2 \log(KN^\star(k)/\delta)}{\Delta_k^2}$ by construction of $e_n$, and this quantity less than $N^\star(K)$ by the supposition. Then, because $K \notin S$ at time $N^\star(K)$, if $N^\star$ is decreasing in $k$, the algorithm will not be forced to terminate at time $N^\star(K)$ by number of epochs, only if all arms other than 1 have already been eliminated, under which the statement would hold anyway. We can then use the same construction for each $k = K - 1, \ldots, \ell$, proving 3. $\qquad\square$

**Lemma 6** (Bound on probability of misidentification on the good event). *For any bandit instance $\nu$ satisfying $\mu_1 \geq \mu_2 \geq \cdots \geq \mu_K$, and $N^\star$ decreasing in $k$, if $M \in \{2, \ldots, K\}$ is the smallest value such that for each $k = M + 1, \ldots, K$, $\Delta_k > \sqrt{\frac{16\sigma^2 \log(KN^\star(k)/\delta)}{N^\star(k)}}$ (if no $\Delta_k$ satisfy this, $M = K$), then $\mathbb{P}_{\nu,\pi}(\{\widehat{I} = j\} \cap G) = 0$ if $j > M$ and $\mathbb{P}_{\nu,\pi}(\{\widehat{I} = j\} \cap G) \leq \exp(-\frac{N^\star(M)\Delta_j^2}{4\sigma^2})$ otherwise.*

*Proof.* We begin with the case $j > M$. By Lemma 5, arm 1 is never eliminated by the algorithm and arms $j, j + 1, \ldots, K$ are eliminated before round $N^\star(j)$. Then, on $G$, DBCARE only terminates

because either $S = \{1\}$ or $n(\tau) = N^\star(|S|) \geq N^\star(j)$, making $\mathbb{P}_{\nu,\pi}(\{\widehat{I} = j\} \cap G) = 0$. Now consider $j \leq M$. Then,

$$
\begin{aligned}
\mathbb{P}_{\nu,\pi}(\{\widehat{I} = j\} \cap G) &= \mathbb{P}_{\nu,\pi}(\{\widehat{I} = j\} \cap G \cap \{j \in S \text{ at } \tau\}) + \mathbb{P}_{\nu,\pi}(\{\widehat{I} = j\} \cap G \cap \{j \notin S \text{ at } \tau\}) \\
&= \mathbb{P}_{\nu,\pi}(\{\widehat{I} = j\} \cap G \cap \{j \in S \text{ at } \tau\}) \\
&= \mathbb{P}_{\nu,\pi}(\{\hat{\mu}_j(n(\tau)) > \hat{\mu}_1(n(\tau))\} \cap G \cap \{j \in S \text{ at } \tau\}) \\
&\leq \mathbb{P}_{\nu,\pi}(\{\hat{\mu}_j(N^\star(M)) > \hat{\mu}_1(N^\star(M))\} \cap G \cap \{j \in S \text{ at } \tau\}) \\
&= \mathbb{P}_{\nu,\pi}\left(\left\{\frac{1}{N^\star(M)} \sum_{n=1}^{N^\star(M)} (Y_{j,n} - Y_{1,n}) > 0\right\} \cap G \cap \{j \in S \text{ at } \tau\}\right) \\
&\leq \mathbb{P}_{\nu,\pi}\left(\frac{1}{N^\star(M)} \sum_{n=1}^{N^\star(M)} (Y_{j,n} - Y_{1,n}) > 0\right) \\
&\leq \exp\left(-\frac{N^\star(M)\Delta_j^2}{4\sigma^2}\right)
\end{aligned}
$$

$\square$

**Lemma 7** (Bound on simple regret on the good event). *For any bandit instance $\nu$ satisfying $\mu_1 \geq \mu_2 \geq \cdots \geq \mu_K$, and $N^\star$ decreasing in $k$, if $M \in \{2, \ldots, K\}$ is the smallest value such that for each $k = M+1, \ldots, K$, $\Delta_k > \sqrt{\frac{16\sigma^2 \log(KN^\star(k)/\delta)}{N^\star(k)}}$ (if no $\Delta_k$ satisfy this, $M = K$), then,*

$$
\mathbb{E}_{\nu,\pi}[(\mu_1 - \mu_{\widehat{I}})\, \mathbb{1}_G] \leq \sqrt{\frac{4\sigma^2}{\sqrt{e}N^\star(M)}}
$$

*Proof.* We begin by relating the simple regret with the probability of misidentification by applying Lemma 6,

$$
\begin{aligned}
\mathbb{E}_{\nu,\pi}[(\mu_1 - \mu_{\widehat{I}})\, \mathbb{1}_G] &= \sum_{i=2}^{K} \Delta_k\, \mathbb{P}_{\nu,\pi}(\{\widehat{I} = k\} \cap G) \\
&\leq \sum_{k=2}^{M} \Delta_k\, \mathbb{P}_{\nu,\pi}\left(\bigcap_{\ell=1}^{k-1} \{\hat{\mu}_k(n(\tau)) > \hat{\mu}_\ell(n(\tau))\} \cap G\right)
\end{aligned}
$$

Now, consider that for $k > \ell \geq 2$, $\mathbb{P}_{\nu,\pi}(\{\hat{\mu}_k(n(\tau)) > \hat{\mu}_\ell(n(\tau))\} \cap G)$ is maximized when $\mu_k = \mu_\ell$ and is equal to $1/2$ when this is the case. Thus, again applying Lemma 6, we can write,

$$
\begin{aligned}
\mathbb{E}_{\nu,\pi}[(\mu_1 - \mu_{\widehat{I}})\, \mathbb{1}_G] &\leq \Delta_2 \sum_{k=2}^{M} \frac{\mathbb{P}_{\nu,\pi}(\{\hat{\mu}_2(n(\tau)) > \hat{\mu}_1(n(\tau))\} \cap G)}{2^{k-1}} \\
&\leq 2\Delta_2\, \mathbb{P}_{\nu,\pi}(\{\hat{\mu}_2(n(\tau)) > \hat{\mu}_1(n(\tau))\} \cap G) \\
&\leq 2\Delta_2 \exp\left(-\frac{N^\star(M)\Delta_2^2}{4\sigma^2}\right)
\end{aligned}
$$

Maximizing in terms of $\Delta_2$ then proves the statement. $\square$

With this collection of technical lemmas providing control on the behavior of DBCARE, we are ready to prove Theorems 2 and 4.

*Proof of Theorem 2.* We break this proof into two cases. First, consider problems of complexity $H \leq (\sigma^2 c)^{-1}$ with $\mu_1 \geq \mu_2 \geq \cdots \geq \mu_K$. Further, let $M \in \{1, \ldots, K\}$ be the smallest value such that for each $k = M+1, \ldots, K$, $\Delta_k > \sqrt{16ek\sigma^2 c \log(KN^\star(k)/\delta)}$ (if no $\Delta_k$ satisfy this, $M = K$). Then, by the definition of $H$, we can write

$$
\mathrm{LB}_{\mathrm{MI}}(H) = \frac{\sigma^2 cH}{4} \log\left(\frac{e}{\sigma^2 cH}\right) \geq \frac{M-1}{64eM \log(KN^\star(M)/\delta)} + \frac{\sigma^2 c}{4} \sum_{k=M+1}^{K} \frac{1}{\Delta_k^2} \tag{12}
$$

Now, if $M \geq 2$, we apply Lemmas 4 and 5 to show the following:

$$\mathbb{P}_{\nu,\pi}(\{\widehat{I} \neq 1\} \cap G) + c\,\mathbb{E}_{\nu,\pi}[\tau\,\mathbb{1}_G] = \mathbb{P}_{\nu,\pi}(\{\widehat{I} \neq 1\} \cap G) + c\sum_{k=1}^{K}\mathbb{E}_{\nu,\pi}[N_k(\tau)\,\mathbb{1}_G]$$

$$\leq 1 + \frac{2\log(M)}{e} + 16\sigma^2 c\sum_{k=M+1}^{K}\frac{\log(KN^\star(k)/\delta)}{\Delta_k^2} \qquad (13)$$

If, in fact, $M = 1$, then combining the results of Lemmas 4, 5, and 6, we can write

$$\mathbb{P}_{\nu,\pi}(\{\widehat{I} \neq 1\} \cap G) + c\,\mathbb{E}_{\nu,\pi}[\tau\,\mathbb{1}_G] \leq 16\sigma^2 c\sum_{k=M+1}^{K}\frac{\log(KN^\star(k)/\delta)}{\Delta_k^2} \qquad (14)$$

Then, multiplying (12) by $760\log(K)\log(K\log(K)/ec^2)$ and adding $Kc$ (to account for non-integer pulls) then upper bounds both (13) and (14). Now, consider the case where $H > (\sigma^2 c)^{-1}$. Then $\mathrm{LB}_{\mathrm{MI}}(H) = 1/4$. Directly applying Lemma 4 gives us for all $H$,

$$\mathbb{P}_{\nu,\pi}(\{\widehat{I} \neq 1\} \cap G) + c\,\mathbb{E}_{\nu,\pi}[\tau\,\mathbb{1}_G] \leq 1 + \frac{2\log(K)}{e} \leq 760\log(K)\log\left(\frac{K\log(K)}{ec^2}\right)\left(\frac{1}{4}\right)$$

Finally, consider that by our choice of $\delta$, using Lemmas 4 and 5, regardless of the value of $H$, we have

$$\mathbb{P}_{\nu,\pi}(G^c)\left(\mathbb{P}_{\nu,\pi}(\widehat{I} \neq 1 \mid G^c) + c\,\mathbb{E}_{\nu,\pi}[\tau \mid G^c]\right) \leq \delta\left(1 + \frac{2\log(K)}{e}\right) \leq c$$

This then proves the statement. $\qquad \square$

*Proof of Theorem 4.* This proof largely mirrors that of 2. Again, first consider problems satisfying $H\Delta_2^{-1} \leq (\sigma^2 c)^{-1}$ with $\mu_1 \geq \mu_2 \geq \cdots \geq \mu_K$, and let $M \in \{1, \ldots, K\}$ be the smallest value such that for each $k = M+1, \ldots, K$, $\Delta_k > \sqrt{(32e/3)((k-1)\sigma^2 c)^{2/3}\log(KN^\star(k)/\delta)}$ (if no $\Delta_k$ satisfy this, $M = K$). Then,

$$\mathrm{LB}_{\mathrm{SR}}(H) \geq \frac{3(M-1)^{1/3}(\sigma^2 c)^{1/3}}{128e\log(KN^\star(M)\delta^{-1})} + \frac{\sigma^2 c}{4}\sum_{k=M+1}^{K}\frac{1}{\Delta_k^2} \qquad (15)$$

If $M \geq 2$, we apply Lemmas 4, 5, and 7 to show

$$\mathbb{E}_{\nu,\pi}[(\mu_1 - \mu_{\widehat{I}})\,\mathbb{1}_G] + c\,\mathbb{E}_{\nu,\pi}[\tau\,\mathbb{1}_G] \leq \sqrt{\frac{8\sqrt{e}}{3}}((M-1)\sigma^2 c)^{1/3} + \frac{3\log(M)}{2}(M\sigma^2 c)^{1/3}$$
$$+ 16\sigma^2 c\sum_{k=2}^{K}\frac{\log(KN^\star(k)\delta^{-1})}{\Delta_k^2} \qquad (16)$$

Additionally, if $M = 1$, then,

$$\mathbb{E}_{\nu,\pi}[(\mu_1 - \mu_{\widehat{I}})\,\mathbb{1}_G] + c\,\mathbb{E}_{\nu,\pi}[\tau\,\mathbb{1}_G] \leq 32\sigma^2 c\sum_{k=2}^{K}\frac{\log(KN^\star(k)\delta^{-1})}{\Delta_k^2} \qquad (17)$$

Then, multiplying (15) by $575\log(K)\log(K\log(K)B\sigma^{5/3}c^{-4/3})$ upper bounds both (16) and (17). Now, for the case where $H\Delta_2^{-1} > (\sigma^2 c)^{-1}$ and for the worst-case comparison, we apply Lemmas 4 and 7 to show for all $H$,

$$\mathbb{E}_{\nu,\pi}[(\mu_1 - \mu_{\widehat{I}})\,\mathbb{1}_G] + c\,\mathbb{E}_{\nu,\pi}[\tau\,\mathbb{1}_G] \leq \sqrt{\frac{8\sqrt{e}}{3}}((K-1)\sigma^2 c)^{1/3} + \frac{3\log(K)}{2}(K\sigma^2 c)^{1/3} \qquad (18)$$

We then have (18) upper bounded by $4\log(K)(K\sigma^2 c)^{1/3}$, and $\mathrm{LB}_{\mathrm{SR}}(H) \geq 0$. We can also upper bound (18) by $20\log(K)\mathrm{LB}_{\mathrm{SR}}^\star$. Finally, we never incur more than an additional $Kc$ risk due to integer pulls, and by choice of $\delta$,

$$\mathbb{P}_{\nu,\pi}(G^c)\left(\mathbb{E}_{\nu,\pi}(\mu_1 - \mu_{\widehat{I}} \mid G^c) + c\,\mathbb{E}_{\nu,\pi}[\tau \mid G^c]\right) \leq \delta\left(B + \frac{3c\log(K)\sigma^{2/3}}{ec^{2/3}}\right) \leq c$$

which proves all statements. $\qquad \square$

Now, despite our 2-arm results being corollaries of their more general $K$-arm counterparts, we are able to provide tighter constants in Corollaries 2.1 and 4.1 by utilizing some more precise techniques that are not generally applicable in the $K$-arm case. For both cases, we apply Lemma 5 in the 2-arm case to identify a $\Delta^\star$ such that, for all $\Delta > \Delta^\star$, the algorithm is guaranteed to identify the optimal arm before reaching $N^\star$ samples per arm on the good event $G$. We then show that we simply need to find a multiplier which makes the lower bound larger than the upper bound at $\Delta^\star$, and this multiplier will work for all other $\Delta$.

*Proof of Corollary 2.1.* We begin by using Lemma 5 to identify $\Delta^\star = \sqrt{32e\sigma^2 c \log\left(\frac{e+1}{(ec)^2}\right)}$, which, combined with Lemma 6, allows us to write,

$$\sup_{\nu \in \mathcal{M}_\Delta} \mathcal{R}_{\mathrm{MI}}(\pi, \nu) \leq \mathrm{UB}_{\mathrm{MI}}(\Delta) := \begin{cases} \exp\left(-\frac{\Delta^2}{8e\sigma^2 c}\right) + \frac{1}{e} + 3c, & \text{if } \Delta \leq \Delta^\star \\ \frac{32\sigma^2 c \log\left(\frac{e+1}{(ec)^2}\right)}{\Delta^2} + 3c, & \text{if } \Delta > \Delta^\star \end{cases} \tag{19}$$

where the additive $3c$ term is to account for integer pulls for each of the 2 arms and an additional $c$ bound for the expected risk on $G^c$. Clearly, for any $a \geq 128 \log\left(\frac{e+1}{(ec)^2}\right)$, (19) is upper bounded by $a\mathrm{LB}_{\mathrm{MI}}(\Delta) + 3c$ for all $\Delta > \Delta^\star$. We then divide our analysis for the remaining $\Delta$ into two cases: when $\Delta \leq \sqrt{e\sigma^2 c}$ and otherwise. First, when $\Delta \leq \sqrt{e\sigma^2 c}$,

$$\mathrm{UB}_{\mathrm{MI}}(\Delta) \leq \frac{e+1}{e} + 3c, \qquad \mathrm{LB}_{\mathrm{MI}}(\Delta) \geq \frac{1}{2e},$$

and so $\mathrm{UB}_{\mathrm{MI}}(\Delta) \leq 8\mathrm{LB}_{\mathrm{MI}}(\Delta)$ for $\Delta \leq \sqrt{e\sigma^2 c}$. Finally, we must consider $\Delta \in (\sqrt{e\sigma^2 c}, \Delta^\star]$. We begin by comparing (19) and (3) at $\Delta^\star$, then we prove that this is sufficient. This gives us,

$$\mathrm{UB}_{\mathrm{MI}}(\Delta^\star) = \left(\frac{(ec)^2}{e+1}\right)^4 + \frac{1}{e} + 3c, \qquad \mathrm{LB}_{\mathrm{MI}}(\Delta^\star) = \frac{\log\left(32e^2 \log\left(\frac{e+1}{(ec)^2}\right)\right)}{128e \log\left(\frac{e+1}{(ec)^2}\right)}$$

Supposing $c < 1/4$,[2] we can see that $\mathrm{UB}_{\mathrm{MI}}(\Delta^\star) \leq 128 \log\left(\frac{e+1}{(ec)^2}\right) \mathrm{LB}_{\mathrm{MI}}(\Delta^\star)$. Finally, we conclude that this is sufficient to prove the statement by showing that $128 \log\left(\frac{e+1}{(ec)^2}\right) \mathrm{LB}_{\mathrm{MI}}(\Delta) - \mathrm{UB}_{\mathrm{MI}}(\Delta)$ is decreasing for $\Delta \in (\sqrt{e\sigma^2 c}, \Delta^\star]$. We show this here:

$$\frac{\partial}{\partial \Delta} a\mathrm{LB}_{\mathrm{MI}}(\Delta) - \mathrm{UB}_{\mathrm{MI}}(\Delta) = -\frac{a\sigma^2 c}{2\Delta^3} \log\left(\frac{\Delta^2}{\sigma^2 c}\right) + \frac{\Delta}{4e\sigma^2 c} \exp\left(-\frac{\Delta^3}{8e\sigma^2 c}\right)$$

$$\leq -\frac{a\sigma^2 c}{2\Delta^3} + \frac{\Delta}{4e\sigma^2 c} \left(\frac{8e\sigma^2 c}{\Delta^2}\right)^2$$

$$= -\frac{a\sigma^2 c}{2\Delta^3} + \frac{16e\sigma^2 c}{\Delta^3},$$

which is $< 0$ when $a > 32e$, which is true for $a = 128 \log\left(\frac{e+1}{(ec)^2}\right)$. Thus, we have proven $\forall \Delta$,

$$128 \log\left(\frac{e+1}{(ec)^2}\right) \mathrm{LB}_{\mathrm{MI}}(\Delta) \geq \mathrm{UB}_{\mathrm{MI}}(\Delta) \geq \sup_{\nu \in \mathcal{M}_\Delta} \mathcal{R}_{\mathrm{MI}}(\pi, \nu)$$

$\square$

*Proof of Corollary 4.1.* We follow the same general proof strategy as in the previous proof. We again apply Lemma 5 to identify $\Delta^\star = (\sigma^2 c)^{1/3} \sqrt{(8e)/3 \log(2N^\star/\delta)}$ and combine it with Lemma 6 to write,

$$\sup_{\nu \in \mathcal{M}_\Delta} \mathcal{R}_{\mathrm{SR}}(\pi, \nu) \leq \mathrm{UB}_{\mathrm{SR}}(\Delta) := \begin{cases} \Delta \exp\left(-\frac{3\Delta^2}{8e(\sigma^2 c)^{2/3}}\right) + \frac{3}{e}(\sigma^2 c)^{1/3} + 3c, & \text{if } \Delta \leq \Delta^\star \\ \frac{32\sigma^2 c \log(2N^\star/\delta)}{\Delta^2} + 3c, & \text{if } \Delta > \Delta^\star \end{cases} \tag{20}$$

---

[2]Previously, we have not put any restriction on the value of $c$, but we have implicitly assumed $c \ll 1$ by the construction of our problem setting. Consider that, under $\mathcal{R}_{\mathrm{MI}}$, if $c \geq 1/4$, one will perform uniformly best on all instances by simply guessing the optimal arm uniformly at random. We do not explicitly account for this behavior in our algorithm construction for simplicity, but it is unrealistic to let $c \geq 1/4$ in practical settings.

First, when $\Delta < (\sigma^2 c)^{1/3}$, clearly $\text{UB}_{\text{SR}}(\Delta) \leq 4\text{LB}_{\text{MI}}(\Delta) + 2(\sigma^2 c)^{1/3} + 3c$ by (20). Then, noting that $\frac{3B\sigma^{4/3}}{c^{5/3}} \geq \frac{2N^\star}{\delta}$, we can clearly see that $\text{UB}_{\text{SR}}(\Delta) \leq 128 \log\left(\frac{3B\sigma^{4/3}}{c^{5/3}}\right) \text{LB}_{\text{SR}}(\Delta) + 3c$ for $\Delta > \Delta^\star$. To prove this same bound for $\Delta \in [(\sigma^2 c)^{1/3}, \Delta^\star]$, we follow the same technique as in the previous proof. First, when $\Delta \in [(\sigma^2 c)^{1/3}, (\sqrt{e}\sigma^2 c)^{1/3}]$,

$$\text{UB}_{\text{SR}}(\Delta) \leq (\sigma^2 c)^{1/3}\left[\exp\left(\frac{1}{6} - \frac{3}{8e^{2/3}}\right) + \frac{3}{e}\right] + 3c, \qquad \text{LB}_{\text{SR}}(\Delta) \geq \frac{(\sigma^2 c)^{1/3}}{4},$$

and thus $\text{UB}_{\text{SR}}(\Delta) \leq 9\text{LB}_{\text{SR}}(\Delta) + 3c$ for $\Delta \in [(\sigma^2 c)^{1/3}, (\sqrt{e}\sigma^2 c)^{1/3}]$. To prove the bound for $\Delta \in ((\sqrt{e}\sigma^2 c)^{1/3}, \Delta^\star]$, we again compare the two at $\Delta^\star$ and then show that the difference between the functions is decreasing in this range of $\Delta$, and thus this is sufficient. At $\Delta^\star$, we have,

$$\text{UB}_{\text{SR}}(\Delta^\star) = \frac{(\sigma^2 c)^{1/3}\sqrt{\frac{32}{3e}\log(2N^\star/\delta)}}{(2N^\star/\delta)^4} + \frac{3}{e}(\sigma^2 c)^{1/3} + 3c \leq \frac{5}{e}(\sigma^2 c)^{1/3} + 3c$$

$$\text{LB}_{\text{SR}}(\Delta^\star) = \frac{3(\sigma^2 c)^{1/3}}{128e\log(2N^\star/\delta)}\log\left(\frac{32e^{5/2}}{3^{3/2}}\log^{3/2}(2N^\star/\delta)\right) \geq \frac{9(\sigma^2 c)^{1/3}}{128e\log(2N^\star/\delta)}$$

Thus, we have $\text{UB}_{\text{SR}}(\Delta^\star) \leq 128 \log\left(\frac{3B\sigma^{4/3}}{c^{5/3}}\right)\text{LB}_{\text{SR}}(\Delta^\star) + 3c$. We then conclude this portion of the proof by showing $128\log\left(\frac{3B\sigma^{4/3}}{c^{5/3}}\right)\text{LB}_{\text{SR}}(\Delta) - \text{UB}_{\text{SR}}(\Delta)$ is decreasing for $\Delta \in ((\sqrt{e}\sigma^2 c)^{1/3}, \Delta^\star]$. We show this here:

$$\frac{\partial}{\partial \Delta} a\text{LB}_{\text{SR}}(\Delta) - \text{UB}_{\text{SR}}(\Delta) = -\frac{a\sigma^2 c}{4\Delta^3}\log\left(\frac{\Delta^6}{e(\sigma^2 c)^2}\right) - \exp\left(-\frac{3\Delta^2}{8e(\sigma^2 c)^{2/3}}\right)\left(1 - \frac{3\Delta^2}{4e(\sigma^2 c)^{2/3}}\right)$$

This is $< 0$ for any $a \geq 0$ when $\Delta \in ((\sqrt{e}\sigma^2 c)^{1/3}, \sqrt{4e/3}(\sigma^2 c)^{1/3})$. When $\Delta \in [\sqrt{4e/3}(\sigma^2 c)^{1/3}, \Delta^\star]$,

$$\frac{\partial}{\partial \Delta} a\text{LB}_{\text{SR}}(\Delta) - \text{UB}_{\text{SR}}(\Delta) \leq -\frac{a\sigma^2 c}{2\Delta^3} + \frac{3\Delta^2}{4e(\sigma^2 c)^{2/3}}\left(\frac{8e(\sigma^2 c)^{2/3}}{3\Delta^2}\right)^{5/2}$$

$$= -\frac{a\sigma^2 c}{2\Delta^3} + \frac{\sigma^2 c}{4\Delta^3}\left(8^{5/2}\left(\frac{e}{3}\right)^{3/2}\right),$$

which is $< 0$ for any $a > 78$, and in particular, $a = 128\log\left(\frac{3B\sigma^{4/3}}{c^{5/3}}\right)$. Now, all that is left to prove is the worst-case comparison with $\text{LB}_{\text{SR}}^\star$. We can show this simply by considering that (20) is maximized at $\Delta = \sqrt{4e/3}(\sigma^2 c)^{1/3}$, where it takes value $(\sqrt{4/3} + 3/e)(\sigma^2 c)^{1/3} + 3c$, which is clearly upper bounded by $9\text{LB}_{\text{SR}}^\star + 3c$. □

## E   Additional Experiments

Here we provide additional $K$-arm experiments. All experiments were performed using a 3.7GHz AMD Ryzen 9 5900X 12-Core processor with 24 GB of memory. Total runtime across all experiments took approximately 7.5 hours, and safeguards were employed to prevent the fixed confidence algorithms from continuing to sample after already severely underperforming the other methods when the sub-optimality gaps were particularly small ($10/c$ total samples allowed).

**K-arm simulations.** We now include a number of additional $K$-arm experiments to demonstrate that our algorithm continues to perform well compared to traditional fixed budget and confidence methods when we move beyond the simple 2-arm case. For all of our $K$-arm experiments, we choose to use Gaussian arms with $\sigma^2 = 1$ for simplicity. We begin with the "1-sparse" setting, where $\mu_1 = \Delta$ and $\mu_k = 0$ for all $k \neq 1$, resulting in $H = (K-1)\Delta^{-2}$, for $\Delta \in [0.05, 2]$ for the probability of misidentification performance penalty and $\Delta \in [0.05, 3]$ for the simple regret performance penalty. We additionally vary $K$ among 8, 16, and 32. For these experiments, we average across $10^4$ runs each with different random seeds. As in § 4, we compare to Sequential Halving [28] for fixed budget and we use an elimination, or "racing," procedure for fixed confidence, with confidence bounds $\sqrt{4\sigma^2 n^{-1}\log(Kn\delta^{-1})}$. To extend to the $K$-arm case, our "low" budget is now $5K$, and our "high" budget is $250K$, which align with our choices of 10 and 500 in the 2-arm case. We still use $\delta = 0.1$

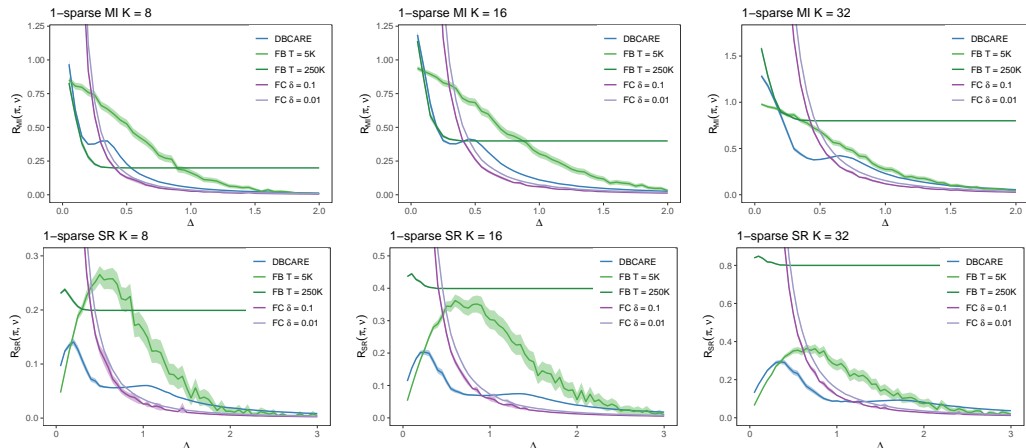

Figure 4: Comparisons between DBCARE and fixed budget and confidence algorithms for $\mathcal{R}_{\mathrm{MI}}$ and $\mathcal{R}_{\mathrm{SR}}$ in the $K$-arm 1-sparse setting. $Y$-axes are adjusted per setting to highlight problem-specific behavior. Confidence regions represent empirical average risk $\pm$ 2 SE.

and $\delta = 0.01$ for our confidences. As we can see in Fig 4, in the 1-sparse setting, DBCARE still enjoys uniformly good performance across the full range of $\Delta$, while the fixed budget and confidence approaches have some region where they perform sub-optimally.

To explore the performance of DBCARE and fixed confidence and budget approaches across a variety of problem structures, we additionally considered the "linear decay" setting, where we set $\mu_1 = \Delta_2$ and $\mu_k = -\Delta_2(\frac{k-2}{K-2})$ for $k \neq 1$ so that the suboptimality gaps linearly increase from $\Delta_2$ to $2\Delta_2$. This results in $H \approx 0.5K\Delta_2^{-2}$. We again let $\Delta_2 \in [0.05, 2]$ for $\mathcal{R}_{\mathrm{MI}}$ and $\Delta_2 \in [0.05, 3]$ for $\mathcal{R}_{\mathrm{SR}}$, average across $10^4$ runs each with a different random seed, and vary $K$ among 8, 16, and 32. As we can see in Fig 5, this setting provides similar results to the 1-sparse and 2-arm settings, with DBCARE performing well across the range of $\Delta_2$ values, while the other methods generally perform sub-optimally for some $\Delta_2$ values.

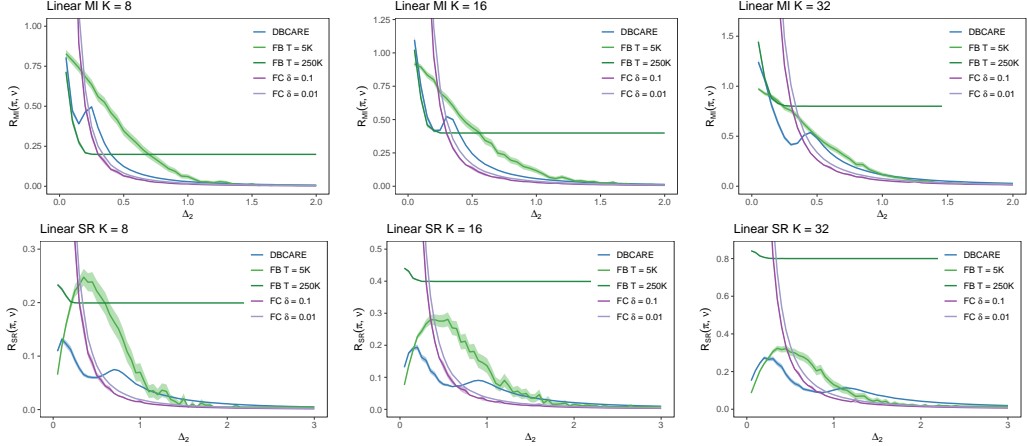

Figure 5: Comparisons between DBCARE and fixed budget and confidence algorithms for $\mathcal{R}_{\mathrm{MI}}$ and $\mathcal{R}_{\mathrm{SR}}$ in the $K$-arm linear decay setting. $Y$-axes are adjusted per setting to highlight problem-specific behavior. Confidence regions represent empirical average risk $\pm$ 2 SE.

