# OpenReview forum: "Balancing Performance and Costs in Best Arm Identification"
_NeurIPS.cc/2025/Conference — NeurIPS 2025 poster_

### Official Review · Reviewer_yixL · 2025-06-15

**Clarity:** 4
**Significance:** 3
**Originality:** 3
**Rating:** 5
**Confidence:** 3

**Summary:**

The authors study how to balance performance and costs in bandit problems.
Specifically, they propose two new settings that interpolate the fixed confidence and fixed budget settings. Specifically, they study:
- A "missidentification setting", where the goal of the agent is minimizing the probability of missidentification + the expected stopping time of sampling multiplied by some costs $c$ which has to be interpreted as the cost of sampling
- A "simple regret" setting, where the goal of the agent is minimizing the distance in mean of the recommended arm wrt to the optimal one + the expected stopping time of sampling multiplied by some costs $c$ which, again, has to be interpreted as the cost of sampling

The authors provide first a detailed description of both problems for the case of $K=2$ arms. Their findings highlight that there exists two regimes: when the cost is large wrt the gap, then, the agent should not sample the arms, but just recommends an arm at random, while, when gap are larges wrt the costs, the agent should explore the arms in order to minimize the two risk functionals that I mentioned above.
Then, the authors extend the results to the K-arm setting and perform some numerical experiments.

Overall, the authors provided lower bounds, and almost matching upper bounds (both in a worst-case sense on the possible problem instances).

**Questions:**

For the K-arm setting I was expecting a smoother transitions in the two regimes according to the gaps of the bandits. For instance, suppose that the top-K/2 arms are very close while the remaining ones are highly sub-optimal. In this case, I was expecting the LB to show that (e.g., for the MI setting) we are paying a fixed error term of proportional to K + a gap dependent quantity that encodes the cost of pulling arms to discriminate the K/2 arms with large gaps.  This is also how the algorithm is somehow working, but e.g., Theorem 2 does not show this sort of problem-dependent behavior.I suspect that this might be related to the weaknesses that I was mentioning above, i.e., the fact that all results are not problem dependent but only problem dependent on the worst-case on the possible instance. Can the authors comment on this point? It seems that this point is also related to the future directions that the authors are mentioning at the end of the work

**Ethical Concerns:**

["NO or VERY MINOR ethics concerns only"]

**Final Justification:**

I am confirming my positive score as I am satisfied with the discussion

**Limitations:**

This work is of theoretical work. I do not foresee a direct path to negative societal impact.

**Paper Formatting Concerns:**

None.

**Quality:**

3

**Strengths And Weaknesses:**

# Strenghts
- Interesting setting that nicely interpolates between fixed budget and fixed confidence setting. It removes the need of specifying an horizon and a confidence level, by just introducing a cost parameter $c$. To the best of my knowledge, this is novel and deserves attention from the community
- The clarity of the exposition is good and the paper is easy to follow.
- The paper highlights nicely the presence of two regimes that depend on costs and gaps. This provides intuition on the challenges of this problem compared to more canonical BAI settings.

# Weaknesses
- All results are not fully problem dependent, but only problem dependent on the worst-case on the possible instance. This raises questions on the tightness of the results (see also my question below). However, I do not believe that this is a sufficient reason to stand for rejection.

# Minor comments
- Proof ideas on the algorithm is rather clear. However, proof ideas for the LB (lines 88-90) is not that informative. I invite the authors to either expand it or remove it in the future
- Additional work on multi-fidelity BAI "Optimal multi-fidelity best-arm identification", Poiani et al., NeurIPS 2024
- As a future direction, I think it would also be interesting to model the cost of pulling an arm in an arm-dependent fashion. Do the authors have any intuition on how (and if) their results extend in this direction?

---

> ### Author Rebuttal · Authors · 2025-07-31
>
> Thank you for your response and questions.
>
> **Weaknesses/Questions**:
> - *Worst-case dependence/Specific settings*: The worst-case performance that we study is the worst-case on a **fixed** gap or complexity, so while these results are not based on specific bandit instances, they are still gap/complexity-dependent results. This is in line with gap- and complexity-dependent bounds elsewhere in the BAI literature, and serves to eliminate pathological cases such as good performance of an algorithm which always returns a particular index, or problems where the noise is so small that just a single sample from each arm is sufficient to identify the optimum.
>   - Regarding the specific \\(K/2\\) close gaps structure you mention, this type of behavior is encoded via the complexity measure \\(H\\) in the lower bound. If \\(K/2\\) of the arms have some small gap \\(\Delta\\) and the remaining have a much larger gap, then the lower bound will be proportional to \\(K\Delta^{-2}\\) plus some much smaller quantity from the large gaps.
> - *Proof ideas*: Given more space in the final version of our work, we are happy to expand upon the lower bound proof idea to increase clarity.
> - *Additional multi-fidelity BAI work*: Thank you for pointing out this publication, we will include this in our remarks on multi-Fidelity BAI in the final version.
> - *Arm-dependent costs*: We are also interested in this as a future direction! We made some preliminary explorations into the effect of heterogeneous costs in the 2-arm setting, and it appears that the role of \\(c\\) in the lower bound would be replaced by \\(c\_1+c\_2\\), while the oracle upper bound depends on \\((\sqrt{c\_1}+\sqrt{c\_2})^2\\). We believe that we could allow \\(N^\star\\) to provide a different maximal number of samples for each arm depending on the cost for that arm, but we thought it prudent to consider a more streamlined approach with a single cost parameter shared by all arms when introducing this new paradigm, especially since heterogeneous costs greatly complicated the analysis. We also believe that this was appropriate for this work because a single cost parameter is relevant in many settings such as some advertisement A/B testing or focus group testing.

---

> ### Comment · Reviewer_yixL · 2025-08-02
> **Positive score confirmed**
>
> I thank the authors for their rebuttal and the other reviewers for their comments.
>
> I am confirming my positive score as I am satisfied with the author's responses.

---

### Official Review · Reviewer_1bsu · 2025-06-26

**Clarity:** 3
**Significance:** 2
**Originality:** 3
**Rating:** 4
**Confidence:** 3

**Summary:**

This paper introduces a novel framework for Best Arm Identification (BAI) in multi-armed bandits that directly balances the trade-off between sampling cost and arm performance. Unlike classical fixed-budget or fixed-confidence setups, the proposed setting minimizes a risk functional that penalizes both incorrect arm identification and the cost of pulling arms. It considers two measures: the sum of misidentification probability and sampling cost, and the sum of simple regret and sampling cost. The authors establish instance-dependent lower bounds for both settings, showing a phase transition depending on problem complexity and cost parameters, which does not appear in fixed-confidence/budget setups. A new algorithm, DBCARE, is proposed and shown to nearly match the lower bounds up to polylogarithmic factors and constants across different regimes. Theoretical analysis is complemented with empirical evaluations on synthetic and real-world datasets.

**Questions:**

- Could the authors elaborate on the role of $\delta$ beyond worst-case guarantees? Is there potential to select $\delta$ in a more data-dependent or adaptive way that improves empirical performance?
- Can the authors provide a practical scenario where this problem formulation can be used? It has mentioned clinical trials. But I suppose in these safety-critical fields, identifying a good arm is quite essential and the cost is not the primary concern.

**Ethical Concerns:**

["NO or VERY MINOR ethics concerns only"]

**Final Justification:**

This paper introduces a novel framework for Best Arm Identification (BAI) in multi-armed bandits that directly balances the trade-off between sampling cost and arm performance. The problem and the proposed algorithm is of interest to the community.

**Limitations:**

yes

**Quality:**

3

**Strengths And Weaknesses:**

**Strengths**
- The introduced cost-aware BAI formulation seems to be new from the canonical formulations.
- The paper presents theoretical results, including lower bounds with clear phase transition behavior and matching upper bounds (up to polylogarithmic factors and constants) for the proposed algorithm across regimes.

**Weaknesses**
- The choice of the confidence parameter $\delta$ plays a crucial role in DBCARE’s performance but lacks thorough discussion. A more detailed analysis of its sensitivity or adaptability to instance-specific features would be valuable.
- DBCARE operates in a round-robin fashion and eliminates arms based on confidence intervals, which may limit its adaptivity. It would be beneficial to explore more adaptive sampling strategies that better focus on promising arms.

---

> ### Author Rebuttal · Authors · 2025-07-31
>
> Thank you for your response and questions.
>
> **Weaknesses/Questions**:
> - *Role of \\(\delta\\)*: The confidence parameter \\(\delta\\) used by DBCARE diverges from that used in traditional BAI approaches, and is **not** a hyperparameter to be chosen by users per instance. We explicitly state the values used for \\(\delta\\) in the statements of Corollary 2.1 and 4.1 and Theorems 2 and 4, and we discuss in lines 218-220, 257-258, 304-305, and 325 that \\(\delta\\) is used to control control the worst-case performance of DBCARE, ensuring that, on the event that the true gaps are not contained within the confidence intervals, we incur no more additional risk than the cost of a single arm pull, \\(c\\). We introduce \\(\delta\\) in this manner to guide intuition for this problem as it relates to traditional BAI approaches, and so that we can generalize a single algorithm for use with both risk functionals, \\(\mathcal{R}\_{\text{MI}}\\) and \\(\mathcal{R}\_{\text{SR}}\\).
> - *Sampling scheme*: With its current sampling scheme, combined with \\(N^\star\\) being a decreasing function of the size of the surviving set, DBCARE is in fact adaptive to the problem setting at hand, quickly eliminating highly suboptimal arms and allocating additional resources to promising candidates. Further, this approach appears to be sufficient, as evidenced by the fact that DBCARE is near optimal on nearly every problem instance, and this sampling scheme aligns with many existing algorithms in the BAI literature, such as Sequential Halving, Successive Rejects, and racing / elimination-style algorithms.
> - *Practical applications*: In lines 23-24 and 39-46, we discuss the example of A/B testing in advertising, where net profit is the primary metric that most business end users would be interested in. In lines 24-25, we also mention the ability to apply this framework to hyperparameter tuning and model selection in instances where model performance and computing resources required to evaluate samples can be explicitly traded off. In Appendix E, where we report the results of some additional experiments, we further discuss the utility of our setting when applied to early-stage animal testing and high-throughput experiments for drug discovery and dose finding where revenue for a pharmacology company is relative to drug efficacy and profit is a metric of interest. In general, BAI is an abstraction to help build intuition for general problem structures and more practical applications can be built on top of the ideas introduced in works such as ours.

---

> > ### Comment · Reviewer_1bsu · 2025-08-03
> >
> > Thank the reviewers for the clarifications! I have increased my evaluation.

---

### Official Review · Reviewer_8fq7 · 2025-06-30

**Clarity:** 2
**Significance:** 3
**Originality:** 3
**Rating:** 5
**Confidence:** 4

**Summary:**

This paper proposes a new performance metric on the topic Best Arm Identification, which combines the total expected pulling times and the failure probability(or simple regret) into the loss function. Targeted on minimizing this new metric, this paper discusses an oracular policy and its further extension DBCARE. This paper also provides theoretical analysis for the algorithm DBCARE. The comparison between the upper and lower bounds suggests that the excellency of the algorithm.

**Questions:**

I wonder whether there exist some conclusions for the hardness of achieving best of both. For example, if an algorithm can achieve relatively good performance in the case of large H, does it mean the algorithm will perform worse in the case of smaller H? This is only an optional question, as I think the minimax bound is well acknowledged in the current community.

Based on my understanding, I believe $c$ can be considered the weights of stopping times, while setting the weight of failing to identify an arm is 1. Please elaborate more on the case when $c$ is very small or $c$ is very large.

Meanwhile, given the known parameter $c$, can we derive performance guarantees on $\Pr(\mu_1\neq \mu_{\hat{I}})$ or $\mathbb{E}[\tau]$? I know the current Theorem 2 already provides a common upper bound for these two terms. Can we do better?

Regarding the numerical experiments, I wonder whether you can compare your results with an anytime algorithm. For example, “Anytime Exploration for Multi-armed Bandits using Confidence Information”, by Jun & Nowak, proposes algorithm ATLUCB. We can take the termination time of DBSARE as the termination time of ATLUCB and compare their failure probability. Do you think this comparison is reasonable? If so, can we expect DBCARE achieve similar or even better numeric performance compared to the ATLUCB?

**Ethical Concerns:**

["NO or VERY MINOR ethics concerns only"]

**Final Justification:**

After interacting with the authors and reading other reviews, I think the paper is interesting and will benefit the community. Hence I increase my score. Nevertheless, the authors shall discuss the limitations of the paper as discussed.

**Limitations:**

Yes

**Quality:**

3

**Strengths And Weaknesses:**

Strengths:
1. This paper proposes a new metric formulation on Best Arm Identification, different with the classical fixed-confidence setting and the fixed-budget setting. This new problem formulation is a great supplement for the academic community.
2. This paper provides theoretical analysis for both upper and lower bounds. The comparison suggests the efficiency of the algorithm design, as it can handle both performance metric MI and SR, with modification on some hyperparameters.

Weakness:
1. As discussed by the authors, the upper and lower performance bounds differ in the case of $H\rightarrow \infty$, under the metric $\mathcal{R}_{SR}$.
2. In the numeric experiment, the proposed algorithm DBCARE cannot outperform others significantly. But it is still reasonable, as it remains reasonably good across all the instances.
3. The authors should also compare their results with this preprint: https://arxiv.org/abs/2409.18909, which is closely related to the authors' models.

---

> ### Author Rebuttal · Authors · 2025-07-31
>
> Thank you for your response and questions.
>
> **Weaknesses**:
> - *Differences when \\(H\to\infty\\)*: Here we address the noted weakness and the question regarding the hardness result. In lines 113-118, we note the reason for the inability to improve the lower bound in a straightforward way, which causes this gap between the lower and upper bound to arise. However, we do conjecture that there is a hardness result of the following form: any algorithm which does not have a priori knowledge of the gap and achieves the minimax optimality under the metric \\(\mathcal{R}_{\rm{SR}}\\) **must** have performance of the same order as our algorithm when \\(H\to\infty\\). The key insight here is that any algorithm that wishes to perform on the order of the current lower bound when \\(H\to\infty\\), must necessarily be willing to guess the optimal arm after receiving very little information. This algorithm can perform well on very complex problems (\\(H\to\infty\\)) and very easy problems (\\(H\to0\\)), but will necessarily perform poorly on problems of moderate complexity, which is the region where the maximum of the lower bound occurs, and so algorithms with this behavior should not be able to achieve the minimax rate. We hoped to include a result of this form in this submission, but were unable to rigorously prove the result. We mention this conjecture in lines 359-360, and we do hope that this problem setting is of interest to the community and that a result of this form can be included in a future work.
> - *DBCARE empirical performance*: We believe that DBCARE’s reasonable performance across all problem instances is exactly its appeal, even if it is unable to outperform every other approach on each problem instance. As seen in Figure 2, specific choices for the budget \\(T\\) or the confidence \\(\delta\\) can perform well on certain problem instances, but no choice of budget or confidence exhibits uniformly good performance across all problems. On the other hand, we can see that DBCARE does in fact exhibit uniformly good performance across all problem instances, which is especially desirable when the problem instance at hand is almost never known to a practitioner. Further, DBCARE even often shows performance close to the best hand-tuned choice for the budget or confidence for a particular instance.
> - *Comparison to Yang et al. (2024)*: To our understanding, we believe that the setting considered by Yang et al. differs from our setting in two meaningful ways. First, the setting in Yang et al. still requires a confidence parameter \\(\delta\\) to be specified a priori and algorithms to be \\(\delta\\)-PAC, whereas our setting considers a risk functional which explicitly trades off between performance and sampling costs without the need for a pre-specified confidence parameter. Second, the equivalent of “sampling costs” considered by Yang et al. is defined using the cumulative regret, whereas our setting considers a separate cost parameter \\(c\\) which is not dependent upon the gaps between the arms, but instead represents an external cost associated with accessing samples from the arms, such as paying focus group participants, promoting advertisements on social media during A/B testing phases, or running costly experiments. In lines 146-148 we do discuss the work of Kanarios et al. (2024), who consider a similar setting to Yang et al., though the sampling costs of Kanarios et al. are separately stochastically observed from the rewards, rather than being induced by the cumulative regret, which is more similar to our setting, but is still distinct due to the use of a pre-specified confidence parameter \\(\delta\\). We thank the reviewers for bringing our attention to this work, and we will cite it and discuss it in our related works section in our final version.
>
> **Questions**:
> - *Regarding the cost \\(c\\)*: Because our risk functional linearly trades off between the performance penalty and the sampling cost, you can view \\(c\\) as the relative weight given to a single sample relative to a unit difference in performance. In all of our lower and upper bounds, we considered the cost \\(c\\) to be fixed and wrote our bounds in terms of either the suboptimality gap \\(\Delta\\), in the 2-arm case, or the complexity \\(H\\), in the K-arm case. However, we could instead consider the gap or complexity to be fixed, and glean some insight on the effect of the cost \\(c\\) by considering the bounds to be functions of \\(c\\) instead. In this light, the phase transitions we identified for the gap and complexity now become phase transitions based on the cost \\(c\\), and we can interpret them intuitively. When the cost \\(c\\) is “large” relative to the gap or the complexity, then we observe the limiting behavior that we pointed out under the small gap or large complexity setting, where the optimal behavior is to guess the optimal arm, as the cost of sampling is too large relative to the increase in performance under further sampling. On the other hand, when the cost is relatively small, then the lower bound and the performance of DBCARE are more dependent on the complexity of the particular problem at hand.
> - *Additional performance guarantees*: In order to derive the results of Theorems 2 and 4 showing DBCARE’s performance under our proposed risk functionals compared to the lower bound, we do have to separately bound the probability of misidentification (for Theorem 2) or the simple regret (for Theorem 4) and the expected number of samples (for both). Thus, we could provide guarantees for the individual terms, but the focus of our results is on the performance under the proposed risk functionals, which linearly trade off the performance penalty with the sampling costs.
> - *Comparison with anytime algorithms*: To the best of our understanding, anytime algorithms like ATLUCB, Doubling SAR, and Doubling SH exhibit successively better performance when allowed to continue running over long time horizons, but guarantees on their performance do not apply until surpassing a number of total samples depending on the problem complexity and possibly a pre-specified confidence parameter. Thus, by naively applying the stopping rule used by DBCARE to one of these algorithms, it is unclear to us how those algorithms would perform, and would likely be highly dependent on the particular problem scenario. This is especially due to the fact that the choice of stopping time for DBCARE and its theoretical guarantees are tied into the specific sampling scheme that is used by DBCARE.

---

### Official Review · Reviewer_VxJA · 2025-07-01

**Clarity:** 4
**Significance:** 3
**Originality:** 3
**Rating:** 5
**Confidence:** 4

**Summary:**

The paper propses a novel formulation of the objective in a multi-arm bandit setting: rather than the common fixed budget or fixed confience settings, a natural objective is proposed that offers a natural tradeoff between the performance and the cost of the arm pulls. For this objective, the paper proves lower bounds and upper bounds that match upto logarithmic factors. The techniques used are the standard machinery of change of measure arguments with a clever adaptive scheme for the algorithm.

**Questions:**

- The baselines are unclear -- Which specific fixed-budget and fixed-confidence algorithms served as baselines in your experiments?
- The end of the experiments section mentions experiments on a drug-discovery dataset. Could you highlight the main takeaways from these results and explain how they demonstrate DBCARE’s superiority over traditional BAI approaches in such a real-world context?

**Ethical Concerns:**

["NO or VERY MINOR ethics concerns only"]

**Limitations:**

yes

**Quality:**

3

**Strengths And Weaknesses:**

Strengths:
- nice formulation which avoids the dilemma of what confidence or budget to set, and instead offers an automatic tradeoff between performance and number of arm pulls.
- nice explanation of the intuition by considering 2 arm case and policyy with known $\Delta$ before doing the general case.
- elegant analysis showing an interesting phase transition between regimes of low and high complexity

Weaknesses:
- very limited experimental evaluation.

---

> ### Author Rebuttal · Authors · 2025-07-31
>
> Thank you for your response and questions.
>
> **Weaknesses**:
>
> - *Limited experimental evaluation*: Because we introduced a novel paradigm for multi-armed bandit problems, we found it prudent to focus more on providing intuition for the problem and clearly demonstrating new theoretical results, which limited the available space for experimental evaluation. We have included some additional synthetic and real K-arm experiments in Appendix E to further demonstrate DBCARE’s utility over classical BAI approaches to this setting. Given the additional page allowance for the final copy, we can move some of these results to the main body to provide more immediate experimental demonstrations.
>
> **Questions**:
> - *BAI Baselines*: In lines 342-343, we note that we compared to Sequential Halving for the fixed budget algorithm and to an elimination algorithm using the optimized stopping rules of Kauffman et al. (2016) for the fixed confidence algorithm. In lines 1038-1040 in Appendix E, we note that for our K-arm experiments we again use Sequential Halving for the fixed budget setting and an elimination algorithm for the fixed confidence setting, in this case using confidence bounds of \\(\sqrt{4\sigma^2n^{-1}\log(Kn\delta^{-1})}\\) for the elimination algorithm, as the optimized bounds of Kauffman et al. only hold in the 2-arm case.
> - *Drug Discovery takeaways*: We provide details of the experimental results for the drug discovery dataset in lines 1053-1074 in Appendix E, but we will provide the takeaways here as well. In this experiment, we used the results reported in Table 2 of Genovese et al. (2013) and evaluated the patient response to different doses of the drug secukinumab via the American College of Rheumatology criteria ACR20, ACR50, and ACR70. The goal of the original study was evaluate the probability of a patient achieving at least ACR20 with each dose, so we used two response models in our experiment: the first using a binary response with probability determined by the proportion of patients achieving at least ACR20 in the original study, and a discrete response corresponding to 0.2 for achieving ACR20, 0.5 for achieving ACR50, 0.7 for achieving ACR70, and 0 for not achieving any criteria, with probabilities determined by the patient proportions in the original study. Again comparing to Sequential Halving and an elimination algorithm, we varied the cost \\(c\\) in our experiments, and we demonstrated that across a range of values for \\(c\\), in both response models, and across both the misidentification and simple regret penalties, none of the traditional BAI methods uniformly outperformed DBCARE. We feel that this is a clear demonstration of the utility of using DBCARE under this problem setting, rather than choosing a traditional BAI method, when the true gaps between the means are unknown. Though DBCARE is not guaranteed to give optimal performance on every single problem instance, we have demonstrated both theoretically and empirically that DBCARE will never perform too much worse than the optimal method, and does not require access to any knowledge that would be unreasonable for a practitioner to know a priori.

---

> > ### Comment · Reviewer_VxJA · 2025-08-03
> >
> > Thanks for pointing out the results on the drug discovery dataset in the appendix and other clarifications. I'll stick with my (positive) score.

---

### Decision · Program_Chairs · 2025-09-17

**Decision:**

Accept (poster)

**Comment:**

his paper studies the best arm identification problem for multi-armed bandits. However, in contrast to the fixed budget or fixed confidence setting, the authors propose a novel criteria that balances correctness with sample complexity, motivated by the difficulty of choosing the budget or confidence level. While the new criteria has its own special constant c to choose, reviewers were content with the setting and felt that the paper would make a strong contribution to the (mature) area.